# Bayesian reconstruction of sea-level and hydroclimates from coastal landform inversion: application to Santa Cruz (US) and Gulf of Corinth

Gino de Gelder[1,2,3], Navid Hedjazian[3], Laurent Husson[1], Thomas Bodin[4] , Anne-Morwenn Pastier[5], Yannick Boucharat[1], Kevin Pedoja[6], Tubagus Solihuddin[2], Sri Y. Cahyarini[2]

[1]ISTerre, IRD, CNRS, Université Grenoble-Alpes, Saint-Martin d'Hères, 38400, France

[2]Res. Group of Paleoclimate & Paleoenvironment, Res. Centr. for Climate and Atmosphere, Res. Org. of Earth Sciences and Maritime, National Research and Innovation Agency, Bandung, 40135, Indonesia

[3]ENS de Lyon, CNRS, LGL-TPE, Universite Claude Bernard Lyon1, Villeurbanne, 69100, France

[4]Instituto de Ciencias del Mar (ICM) - CSIC, Barcelona, Spain

[5]GeoForschungsZentrum, Potsdam, 14473, Germany

[6]Université de Caen Normandie, Caen, 14000, France

*Correspondence to*: Gino de Gelder (gino.de-gelder@univ-grenoble-alpes.fr)

**Abstract.** Quantifying Quaternary sea-level changes and hydroclimatic conditions is an important challenge given their intricate relation with paleo-climate, ice-sheets and geodynamics. The world's coastlines provide an enormous geomorphologic archive, from which forward landscape evolution modelling studies have shown their potential to unravel paleo sea-levels, albeit at the cost of assumptions on the genesis of these landforms. We take a next step, by applying a Bayesian approach to jointly invert the geometries of multiple coastal terrace sequences to paleo sea- and lake level variations and extract past hydroclimatic conditions. Using a Markov chain Monte Carlo sampling method, we first test our approach on synthetic marine terrace profiles as proof of concept and then benchmark our model on an observed marine terrace sequence in Santa Cruz (US). We successfully reproduce observed sequence morphologies and simultaneously obtain probabilistic estimates for past sea-level variations, as well as for other model parameters such as uplift and erosion rates. When applied to the semi-isolated Gulf of Corinth (Greece), our method allows deciphering the geomorphic Rosetta stone at an unprecedented resolution, revealing the connectivity between the Lake/Gulf of Corinth and the open sea for different hydroclimatic conditions. Eustatic sea-level and changing sill depths drive marine and transitional phases during interglacial and interstadial periods, whereas wetter and drier hydroclimates respectively over- and under-fill Lake Corinth during interstadial and glacial periods.

**Short summary.** Marine terrace sequences - staircase-shaped coastal landforms - record sea-level changes, vertical motions and erosional processes that are difficult to untangle. To do so we developed a numerical inversion approach: using the observed landscape as input, we constrain the ensemble of parameter ranges that could have created this landscape. We apply the model to marine terrace sequences in Santa Cruz (US) and Corinth (Greece) to reveal past sea/lake levels, uplift rates and hydro-climates.

# 1 Introduction

Reconstructions of Quaternary sea-level variations provide crucial constraints on thresholds and feedbacks within climatic and geodynamic systems that help understand how contemporary climate change may affect future sea level (Lambeck and Chappell, 2001; Hay et al., 2014; Dutton et al., 2015; Shakun et al,. 2015; Austermann et al,. 2017). A key archive of past sea-level is exposed within the geomorphology of most of the world's coastal areas in the form of paleo-shorelines (Johnson and Libbey, 1997; Pedoja et al., 2011, 2014; Rovere et al., 2023; Fig. 1a), but it remains difficult to accurately translate coastal observations and measurements into paleo-sea-level estimates, and to evaluate the uncertainties inherent to these conversions. Major challenges include 1) dating of these landforms, as most paleo-shorelines are erosive in nature (Pedoja et al., 2014) and absolute dating techniques themselves are complex and prone to large uncertainties (Strobl et al., 2014; Hibbert et al., 2016; Ott et al., 2019), 2) observational bias, which are mostly restricted to the most recent glacial cycle(s) and to periods where relative sea level was at similar elevations to present-day (Medina-Elizalde, 2013; Hibbert et al., 2016), 3) absence of reciprocity between paleo-shorelines and sea-level stands, as not all highstands lead to paleo-shorelines, and paleo-shorelines may have formed during one or many sea-level cycles (Guilcher, 1974; Malatesta et al., 2021; Chauveau et al., 2023), and 4) separating the tectonic from the sea-level component within relative sea-level changes (Pedoja et al., 2011). Numerical models of landscape evolution started to overcome some of these limitations, by providing a means to quantitatively interpret undated paleo-shorelines, incorporate full sea-level curves instead of highstands only, unravelling the creation of paleo-shorelines formed over multiple glacial cycles, and considering multiple sea-level curves (e.g. Webster et al., 2007; Jara-Muñoz et al., 2019; Leclerc and Feuillet, 2019; De Gelder et al., 2020; 2023). So far, such numerical models have mainly been used for forward modelling approaches, where a number of proposed sea-level curves are used to predict shorelines, which are then compared to actual observations. However, this only provides a limited way to explore the full ensemble of possible sea-level histories and other model parameters like rock erosion rates or effective wave base depths, which are difficult to estimate. It follows that uncertainties in sea-level estimates from marine terraces remain poorly known, regardless of the method used, and in spite of uniformization attempts (Lorscheid and Rovere, 2019).

Semi-isolated marine basins, i.e. bodies of water that have been connected to the open sea in some intervals of their geologic history, and little or disconnected from the sea in other intervals, develop in hydrodynamic settings for which it is particularly complex to reconstruct sea and lake level. Such basins, like the Red Sea, Sea of Marmara (Turkey), Carioco Basin (Venezuela) and Gulf of Corinth (Greece), have a special geologic interest, given the active tectono-sedimentary processes driving their formation (e.g. Van Daele et al., 2011; McNeill et al., 2019), their sensitivity to rapid sea-level and climatic changes (e.g. Aksu et al., 1999; Siddall et al., 2004), and their role in dispersion of species (e.g. Derricourt, 2005). The main complexity in reconstructing sea-/lake-level fluctuations in such settings, is that 1) during disconnected phases these basins may have been underfilled or overfilled depending on local hydroclimate, and 2) the structural highs (sills) separating the basins from the sea can be simultaneously affected by tectonic vertical motion, sedimentation and erosion.

In this study, we intend to overcome common marine terrace analysis limitations, by using a Bayesian approach to invert the geometry of paleo-shoreline sequences. Our approach provides probabilistic estimates of paleo sea-level, erosion rates, uplift rates, wave-based depths and initial slopes. We focus on erosive marine terraces (Fig. 1b), which are both the most common

type of paleo-shoreline (Pedoja et al., 2014), and are simpler to model than their depositional and bio-constructed equivalents (e.g. Pastier et al., 2019). We first apply our probabilistic inversion approach to a set of synthetic coastal profiles to test and illustrate the method, after which we invert a well-studied marine terrace sequence in Santa Cruz (US) to benchmark our model on a natural example. Finally, we use our approach on the semi-isolated Gulf of Corinth to decipher the complex combination of tectonic uplift, sea- and lake-level fluctuations, local climatic drivers and sill dynamics. The Gulf of Corinth has been considered as a natural rift laboratory and has therefore received a lot of attention as a primary example for studying tectonic and surface processes in young rift systems (e.g. Armijo et al., 1996; Bernard et al., 2006; Nixon et al., 2016). As a semi-isolated marine basin it has also been subjected to many studies on its environmental and climatic evolution (e.g. Collier et al., 2000; Watkins et al., 2019), yet, so far, previous studies could not resolve to what extent sea and lake levels have fluctuated.

These case studies highlight how we can derive probabilistic estimates of past sea-level from marine terraces, and how the natural archive of paleo-shorelines can be further utilized to improve both paleo sea-level estimates, and unravel complex tectono-hydro-climatic interactions.

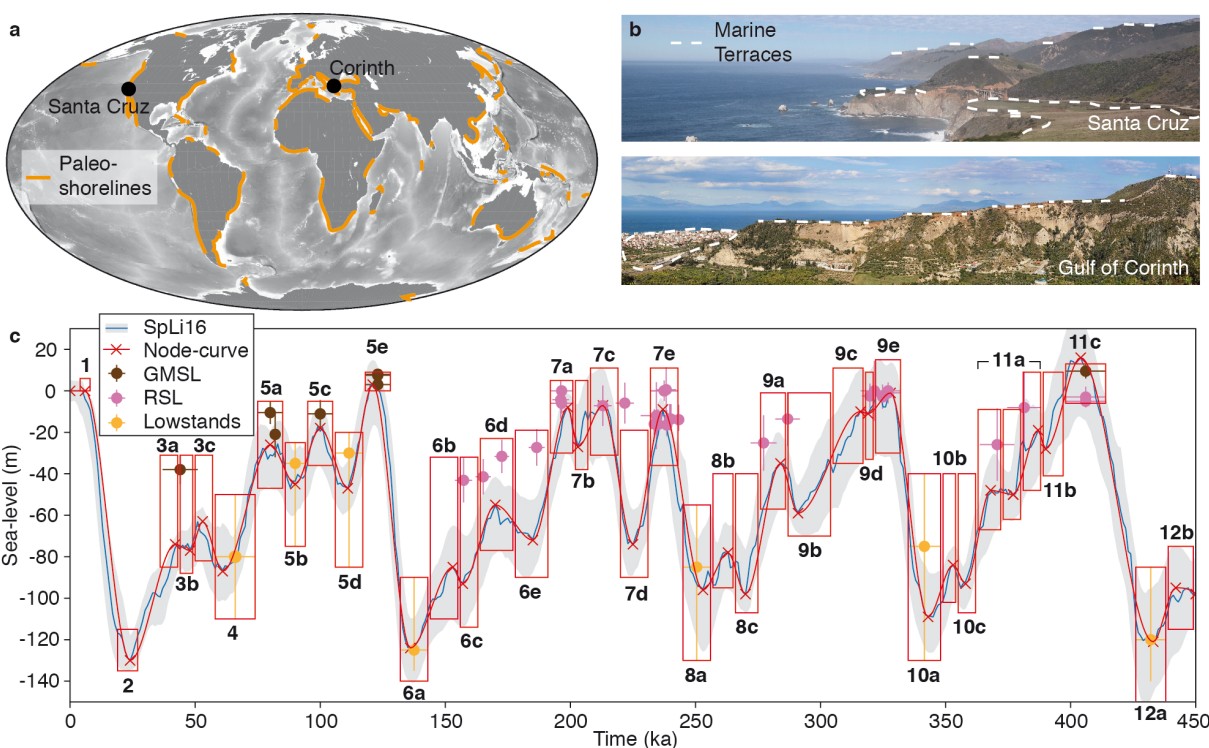

**Figure 1. Paleo-shorelines and paleo-sea level. a)** Global compilation of paleoshoreline sequences, adjusted from Pedoja et al., 2014, **b)** pictures of marine terrace sequences from Santa Cruz (US) and the Corinth Rift (Greece) and **c)** paleo-sea-level estimates for the past 450 ka, showing a sea-level curve (SpLi16, blue; Spratt and Lisiecki, 2016) derived from principal component analysis of 7 sea-level curves with its 2.5% and 97.5% likelihood range (grey envelope), an approximation of that curve with nodes and a cubic spline interpolation (red), global mean sea-level highstand estimates adjusted for glacio-istostatic adjustments (GMSL, brown dots; Kopp et al., 2009; Dutton et al., 2015; Pico et al., 2016; Creveling et al., 2017; Dyer et al., 2021; Tawil-Morsink et al,. 2022), selected relative sea-level highstand estimates >130 ka (RSL, pink dots; Stirling et al., 2001; Murray-Wallace, 2002; Andersen et al., 2010; de Gelder et al., 2022; Marra et al.,

2023), global mean sea-level lowstand estimates from ice sheet data (orange dots; Batchelor et al., 2019), and red boxes that represent the likely admissible range of relative sea-level elevations at locations far from the major ice-sheets (details in Supplementary Information and Table S1) that we consider in this study. Marine Isotope Stages (MIS) are given in bold, and based on Railsback et al., 2015.

## 2 Marine terrace sequence inversion

Marine terraces are relatively flat surfaces of coastal origin, either horizontal or gently inclined seawards (Fig 1b; Pirazzoli, 2005). They are bounded inland by a fossil sea-cliff, and can be covered by a layer of coastal sediments. Here we model erosive marine terraces, which are primarily formed by sea-cliff retreat in response to wave action. The superposition of Quaternary sea-level variations (Fig. 1c) and vertical land movement typically leads to a staircase landscape exhibiting marine terraces sequences (Fig. 1b). In the following sub-sections, we describe the inversion of marine terraces in 4 parts: 1) the unknown model parameters and their relation to observed terraces, 2) the Bayesian formulation of the inverse problem, 3) the Monte Carlo algorithm to approximate the probabilistic solution and 4) the bounds of the uniform prior ranges.

### 2.1 Model parameters

As a general formulation, we can consider:

$$d = g(m) + \varepsilon \qquad (1)$$

Where $d$ describes the vector of observations, in our case the topographic profile of a terrace sequence, and $m$ is the set of unknown model parameters to be inverted for: uplift rate ($U$), erosion rate ($E^*$), wave-base depth ($z_0$), initial slope ($\alpha$) and sea-level history described by a set of nodes (see below). The function $g$ describes the numerical erosion model that links these model parameters to the topographic profile, here the REEF code (Husson et al., 2018; Pastier et al., 2019). Data errors here are given by a random variable $\varepsilon$ that describes the inability of the forward model $g(m)$ to explain the observations $d$.

REEF is a nonlinear model, in which wave erosion is based on the wave energy dissipation model developed by Anderson et al. (1999). The model assumes that the vertical seabed erosion rate is a linear function of the rate of wave energy dissipation against the seabed (Sunamura, 1992). Horizontal erosion rates depend on the energy available at the sea-cliff after dissipation of the far-field wave energy (Anderson et al., 1999). The dissipation rate is dictated by the water depth profile, which increases landwards exponentially with decreasing water depth. The initial erosion rate $E^*$ is expressed as an effective eroded volume per unit of time and coastal length, in which a fraction of erosional residual power $E_r$ erodes the foundation at each location along the curvilinear profile $s$. This fraction $E_r$ depends on the true local water depth along the profile $h(s)$, the water depth for wave base erosion $z_0$, and a coefficient for sea bed erodibility $K$, so that:

$$\frac{\partial E_r}{\partial t} = K \times E_r \times exp(-\frac{h(s)}{z_0}) \qquad (2)$$

Then a residual power $E^* - \int_s \frac{\partial E_r}{\partial t} ds$ carves out a 1 m high notch and all its overhanging material to form a cliff.

Following previous studies (Anderson et al., 1999; Pastier et al., 2019), for K we use 0.1 as bedrock erodibility and 1 for notch carving. Finally, the 2D model we use consists of a landmass with a seaward dipping linear initial slope ($\alpha$; Fig. 2a), an

initial erosion rate ($E^*$; Fig. 2a) that evolves as platforms are being carved, a wave base depth ($z_0$; Fig. 2a) that determines the vertical range over which erosion takes place, a land uplift rate ($U$; Fig. 2a) and a sea-level history.

To invert the morphology of the marine terrace sequences, we parameterize sea-level history (*sl*) with a finite number of unknown parameters. We use nodes interpolated through a cubic spline scheme (Fig. 2b; light blue), in which each node has values for age and elevation.

This creates sea-level curves with similar characteristics to published sea-level curves (red line, Fig. 1c), in which the nodes represent sea-level minima (lowstands) and maxima (highstands) that are typically linked to even and odd-numbered marine isotope stages (MIS), respectively.

*2.2 Bayesian formulation of the inverse problem*

Reconstructing the sea-level history from present day observations of marine terrace sequences can be mathematically formulated as a highly non-linear inverse problem where the solution is non-unique. To embrace this non-uniqueness, the problem can be cast in a Bayesian (probabilistic) framework where the solution is a posterior probability distribution describing the probability of the model parameters (*m*), given the observed data (*d*).

One benefit of Bayesian inference is the ability to propagate uncertainty estimates from the observed measurements ($\varepsilon$ in eq. 1) towards the unknown model parameters. For that, a likelihood probability distribution needs to be defined, based on the mathematical model for uncertainty estimates associated with observations. The errors $\varepsilon$ about the observed shorelines account for the inability of our numerical model *g(m)* to explain observations *d*, and can be due to both observational (either instrumental or due to a naturally degrading morphology) or modelling errors. These errors are supposed random and normally distributed. Their statistics can therefore be described by a Covariance matrix $C_d$ defined by a standard deviation ($\sigma$; Fig. 2a) and a level of spatial correlation (*corrl*; Fig 2a). In this way, the likelihood distribution defining the probability of observing the data *d*, given a set of model parameters *m*, writes:

$$p(d\,|\,m) \propto exp(-\frac{1}{2}[d - g(m)]^T C_d^{-1}[d - g(m)]) \qquad (3)$$

In order to probabilistically estimate the model parameters, Bayes' theorem can be used to combine this likelihood distribution (information given by the data), with the prior distribution *p(m)* that represents independent information on model parameters.

$$p(m\,|\,d) \propto p(d\,|\,m) \times p(m) \qquad (4)$$

The solution of the inverse problem *p(m|d)* is the posterior solution and represents the probability of the model parameters given the data. The prior distribution for each parameter is defined as a uniform distribution between two bounds. That is, we assign a priori equal probability to all the values within the specified range. For example, the position of the nodes defining the sea level history can take a priori any value within the red boxes in Figure 2.

In this work, the data vector is defined as the horizontal position of a set of points measured on the shoreline with a vertical step size (*ipstep*; Fig. 2a). That is, the observed profile is defined as *d($z_i$),* where $z_i$ (with *i=1,2,..,N*) is a regular grid of elevations. The misfit between this observed topography and the modelled paleo-shoreline sequences is therefore measured

by comparing the horizontal distance between observed and simulated horizontal position at each vertical grid point $z_i$. Note that the misfit can also be calculated along vertical axis (i.e. comparing elevations between observed and estimated profiles). We tested this (Fig. S1), but found it harder to reproduce realistic marine terrace sequences with a few m of terrace height variability, and a few hundred meter of terrace width variability (e.g. Regard et al., 2017; De Gelder et al., 2020). Regardless, we note that changing the misfit calculation from horizontal to vertical misfit does not change the paleo sea-level posterior distribution (Fig. S1).

*2.3 Sampling the posterior solution*

We use a Markov chain Monte Carlo algorithm to sample the posterior distribution and explore the range of models that can explain the observed topography within errors. For a review of Bayesian inference and Monte Carlo methods in the geosciences, we refer the reader to Mosegaard and Sambridge (2002), and Gallagher et al. (2009). In the Monte Carlo exploration of the model space, nodes can either be fixed at certain ages and elevations, or treated as unknown parameters, and left free to move within a prescribed range (e.g. red boxes in Fig. 2b,d,e). The 4 main erosion model parameters ($\alpha$, $E^*$, $z_0$, $U$) can also be fixed to chosen values, as done in the synthetic tests below, or left free within chosen ranges, as done for the Santa Cruz and Corinth examples below. The algorithm samples the parameter space as a random walk, where at each iteration a new sea level history model is proposed by perturbing the current one. The proposed model is then either accepted or rejected following an acceptance rule based on the level of data fit of the current and proposed models. The final solution is a large ensemble of paleo sea-level models that approximates the probabilistic solution. That is, the distribution of models follows the posterior probability solution. Of the 1 million model runs per figure, the first 50 000 models were discarded as burn-in models. To verify that the random walk samples the target distribution, we show a number of convergence diagnostics in Fig. S2, which includes parameter acceptance ratios and likelihood evolution for all model runs in this article.

*2.4 Bounds of the uniform prior*

For the different unknown model parameters, imposed prior constraints can be either restrictive, open or anything in between. Within the synthetic tests (section 3) we left the sea-level nodes open, and the other parameters either fixed (Fig. 2) or open (Fig. S3). For the Santa Cruz benchmark tests (section 4) we left the erosion rate, wave base depth and initial slope parameters relatively open, but placed a more restrictive range on the uplift rate of 1.3-1.65 mm/yr, so that the chronostratigraphy of the modeled terrace sequence would match published ages (Perg et al., 2001). We put soft prior constraints on the sea-level history, by restricting possible solutions to the red boxes in Fig. 1c, which represents a cautious interpretation of the ensemble of previous sea-level studies. We adopted a similar strategy for the Corinth terraces (section 5), with the erosion rate, wave base depth and initial slope parameters left relatively open, but stronger prior constraints on uplift rate ranges to respect previous findings on the chronostratigraphy. We tested models with the sea-/lake-level nodes either completely open between 15 and -150 meter elevation (Fig. S4), and with stronger prior constraints only on the highstands. During these highstands the Gulf of Corinth undoubtedly experienced marine conditions (McNeill et al., 2019), so the sea-level elevations can be expected to fall within the red boxes of eustatic sea-level defined in Fig. 1c, whereas lowstands are likely lacustrine.

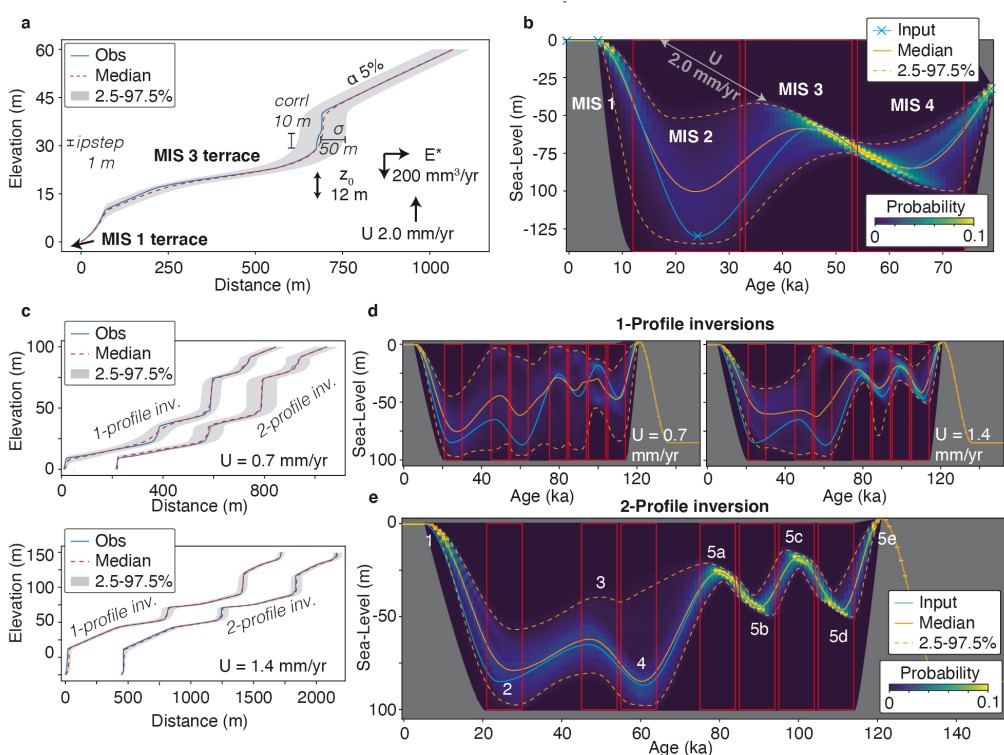

**Figure 2. Inversion of synthetic marine terrace profiles. a)** Synthetic topography (Obs, blue) created from a forward model with known input parameters: $\alpha$ = Initial Slope, $E^*$ = Erosion Rate, $U$ = Uplift Rate and $z_0$ = wave-base depth. The posterior range of models that fit the observed topography with the given $\sigma$, *ipstep* and *corrl* (see text) is represented by the median (orange), and the 2.5 and 97.5 percentiles of the inverted models (grey envelope). **b)** Posterior probability distribution for the sea-level histories. Each individual paleo sea-level history is described with 6 sea-level nodes linked with a cubic spline interpolation, of which the nodes at 78, 6 and 0 ka are fixed in time and elevation, and the other three nodes can move within the three red boxes. The input (target) sea-level history is given in light blue, and the probabilistic solution is depicted by the median (solid orange line) and the 2.5 and 97.5 percentiles (dashed orange lines). **c)** Same as **a**, but with different uplift rates and sea-level histories. **d)** Sea-level histories for the inverted profiles in **c**, similar to **b** but with a different input sea-level history including more nodes. **e)** Similar to **d**, but inverting the two profiles simultaneously to find a common sea-level curve explaining both profiles. MIS are marked in white.

## 3 Synthetic marine terrace profiles

To test and illustrate the potential of the inversion approach, we inverted synthetic topographic profiles that were produced by forward models with known input parameters (Fig. 2). To start with a relatively short and simple sea-level range, we defined an 80 ka sea-level history consisting of 6 nodes (Fig. 2b; light blue). For the inversion, we fixed the elevation and ages of the nodes at 0, 0 and -30 m and 78, 6 and 0 ka, respectively, and the positions of the other three nodes were left as unknown model parameters to be recovered. In the Monte Carlo exploration of the model space, these three nodes were left free to move within a prescribed range (red boxes in Fig. 2b). All other erosion model parameters ($\alpha$, , $E^*$, $z_0$, $U$; Fig. 2a) were fixed during the inversion at the values used to produce the observed topographic profile. The parameters $\sigma$, *ipstep* and *corrl* were set at 50, 1 and 10 m, respectively. We inverted the topographic profile between 0 and 60 m elevation by sampling

the parameter space with 1 million forward simulations. The solution is a large ensemble of sea-level histories that reflect the probability of the paleo sea-level, given the synthetic coastline topography.

The resulting profiles show an MIS 3 terrace at an elevation range of ~15-30 m (Fig. 2a), whereas an MIS 1 terrace lies below the present-day sea level, and is thus not considered in the inversion. As such, the range of sea-level histories that could have created the MIS 3 terrace is narrower than for the MIS 1 terrace (Fig. 2b). This range is particularly limited for the period of sea-level rise leading up to the MIS 3 peak, suggesting that uplifted marine terraces are more likely to form during periods of relative sea-level rise. This is theoretically expected, as erosion scales with the total duration of sea-level occupation (Malatesta et al., 2021), and simultaneous sea-level rise and land uplift implies favorable conditions for the formation of marine terraces. Another notable feature is the distribution of possible sea-level histories along a diagonal line that corresponds to the uplift rate. This line would reach the maximum terrace elevation when extrapolated to t=0 ka, in line with classic graphical methods (Bloom and Yonekura, 1990). Although the MIS 1 terrace is not inverted, there are some limitations to the magnitude and rate of sea-level rise between MIS 2 and MIS 1 (Fig. 2b), probably because this period determines how much of the MIS 3 terrace is eroded at its distal edge.

For the inversion of every individual profile there should be a trade-off between younger, higher sea-level peaks and older, lower sea-level peaks in line with the fixed uplift rate (as in Fig. 2b). These trade-off effects can be overcome through the joint inversion of multiple profiles with different uplift rates, reducing the uncertainty in sea-level reconstructions. To show this, we also inverted two different topographic profiles produced with different fixed uplift rates but with the same sea-level history over a 135 ka timescale (the last glacial-interglacial cycle; Fig. 2c-e). When the two profiles are inverted individually, the range of possible sea-level histories is relatively wide, and again the sea-level peaks would follow a diagonal line parallel to the uplift rate (Fig. 2c, d). However, if we jointly invert both profiles, i.e. assuming that a unique sea-level history would have created both marine terrace staircase morphologies, the probability distribution for past sea-level narrows, and the median sea-level of the inversion better approximates the input curve (Fig. 2e). The range is particularly narrow for the transgressions leading up to the MIS 5a and 5c highstands, for which the corresponding terraces are well developed in the topographic profiles (Fig. 2c). Similar to the MIS 1 terrace in Fig. 2a, the MIS 1 and 3 terraces in Fig. 2c would be located below sea level for the given parameters, and thus the possible sea-level range is wider for the transgressions leading up to MIS 1 and 3 (Fig. 2d). Also for these highstands though, the sea-level is better constrained for the joint inversion (Fig. 2e) than with the individual inversions (Fig. 2d). This suggests that jointly inverting more profiles would increase even further our ability to constrain sea-level histories. To understand what happens in scenarios in which more parameters are unknown, we repeated the same tests (from Figures 2b-d) with broad prior ranges for the uplift rate, initial slope, erosion rate and wave base height (Fig. S3). Also in this case the joint inversion provides much narrower posterior ranges for paleo sea-level compared to the two individual profile inversions, and not too different from the cases with fixed model parameters (Figs. 2b-d). The posterior ranges for all parameters are consistently smaller for the joint inversions, compared to the individual inversions (Fig. S3).

These synthetic tests imply that in natural examples, sea-level reconstruction should also benefit from the inversion of multiple marine terrace profiles if conditions change between those profiles. In this example we used two different uplift rates for the joint inversion, which lead to a range in different terrace sequence morphologies (Fig. 2c), but an approach

where all parameters, including wave base, erosion rate or initial slope, are undefined a priori (or only within a given range), should lead to a more realistic range of possible sea-level histories.

To put this method to the test in real cases, we selected two well-documented yet contrasting cases, Santa Cruz and the Gulf of Corinth, each having their peculiarities that make them ideal to study the inversion of marine terraces. Santa Cruz is well documented and possible parametric windows are quite narrow, making it an ideal benchmark site for our method. On the other hand, the staircase sequence in the Gulf of Corinth, while equally well documented, is particularly relevant to take advantage of the predictive capacities of our method. There, we can decipher the complex interplay between vertical land motion and sea-/lake-level fluctuations in a semi-closed basin, and better reconstruct its hydrodynamic history where paleoclimatic data are insufficient.

## 4 Santa Cruz marine terrace sequence inversion

The marine terraces along the Santa Cruz coastline (Central California, US) formed through a combination of Quaternary sea-level oscillations and tectonic uplift by nearby active faults (e.g. Bradley, 1957; Anderson and Menking, 1994; Anderson et al., 1999; Perg et al., 2001; Matsumoto et al., 2022). We invert a topographic profile from Rosenbloom and Anderson (1994), who distinguished the original eroded bedrock surface, which we use, from its overlying colluvium for 5 marine terraces. We followed the age interpretation of Perg et al. (2001), suggesting these terraces were formed, from bottom to top, during MIS 1, 3, 5a, 5c and 5e. Unlike in the synthetic tests, here we left uplift rate, erosion rate, wave base depth and initial slope parameters free within a prior range of values. We use the elevation (~170 m) and age (~118-128 ka) of the upper terrace to derive a range of possible uplift rates (1.3-1.65 mm/yr), and simultaneously consider ranges for initial slope (5-15%), wave base depth (1-10 m) and erosion rates (100-800 mm³/yr) in the terrace inversion. We use the same inversion parameters as for the synthetic tests, running 1 million models over 450 ka with the sea-level high- and lowstands limited to the red boxes in Fig. 1 (See Supplementary Information and Table S1). We tested different inversion parameters (Fig. S5), and settled on values of 1 m *ipstep*, 100 m $\sigma$ and 10 m *corrl*, as for these values the corresponding 95% posterior ranges for the variability in terrace height (~5 m; Fig. S5) and width (~150 m; Fig. S5), are along the same order of magnitude as the variability in a swath profile across the Santa Cruz terraces (Fig. S6). Higher values for $\sigma$ and *corrl* result in both broader ranges for accepted terrace heights/widths, and consequently also in broader posterior ranges for paleo sea-level, although we note that the first order shape of the paleo sea-level curves remains similar irrespective of the tested *ipstep*, $\sigma$ and *corrl* values (Fig. S5).

The sampled sea-level histories successfully reproduce the terrace morphology, as evidenced by the low misfit of 2 m (Fig. 3a). As with the synthetic tests (Fig. 2), periods of sea-level rise are better constrained than periods of sea-level fall, and highstands better constrained than lowstands (Fig. 3b). Also here there is a trade-off in sea-level peaks, in which younger, higher sea-level peaks could result in similar shaped marine terraces as older, lower sea-level peaks (e.g. for MIS 3c, Fig. 3e). The models limit the uplift rate to ~1.35-1.6 mm/yr, the initial slope to ~7-9.5%, the wave base depth to 4-10 m and the erosion rate to 200-800 mm/yr (Figs. 3c,d). Notably there is a positive correlation between wave base depth and erosion rate

(Fig. 3d), suggesting a higher value for wave base depth would require a higher erosion rate to create the same marine terrace sequence morphology.

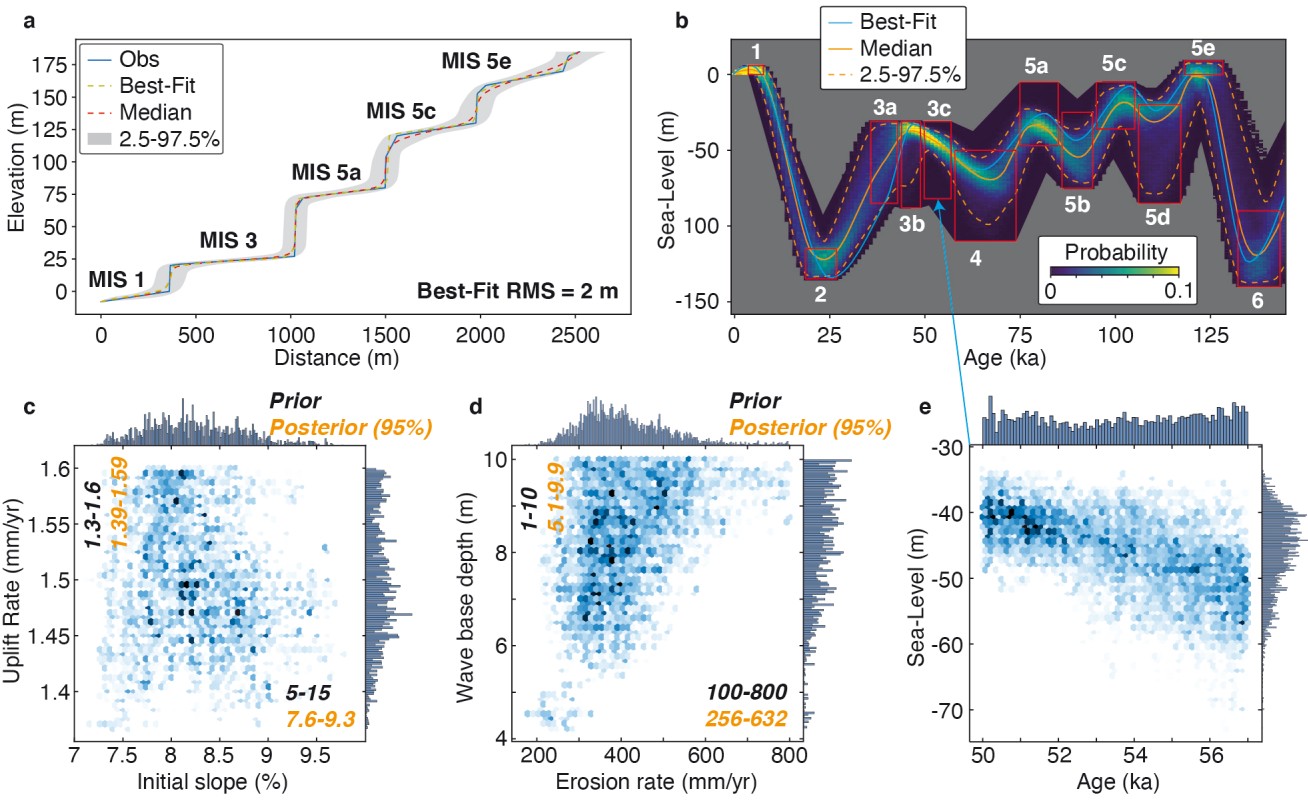

**Figure 3. Inversion of NW-Santa Cruz marine terrace sequence. a)** Observed topography (from Rosenbloom and Anderson, 1994; Obs, blue) with the age interpretation of Perg et al. (2001) marked in bold, together with the modeled best-fit, median, 2.5% and 97.5% percentile profiles. **b)** Probabilistic sea-level reconstruction for the profiles in **a**, MIS in white **c)** Posterior ranges for uplift rates and initial slopes (histogram of the sampled models), **d)** Posterior ranges for wave base depths and erosion rates **e)** Posterior ranges for the MIS 3c peak, i.e. distribution for the position of the 50-57 ka node within the paleo sea level curve.

Compared to our proposed prior range of possible sea-level elevations for MIS 3 (-30 to -80 m; Fig. 3b), the posterior distribution of the inversion suggests paleo sea-level values on the higher end of that spectrum. This is in agreement with a growing number of studies suggesting oxygen-isotope derived sea-level curves underestimate sea-level for that period (Pico et al., 2016; Dalton et al., 2019, 2022; Gowan et al., 2021; De Gelder et al., 2022). For MIS 5a on the other hand, the posterior distribution of the inversion suggests a sea-level peak on the lower end of our proposed range of sea-level elevations (Fig. 3b). Although the highstand posterior distributions still span a broad elevation range of ~25 m, these results tend to align with studies proposing an overall decrease in sea-level between MIS 5e, 5c and 5a (e.g. Chappell and Shackleton, 1986; Schellmann and Radtke, 2004; Tawil-Morsink et al., 2022).

We also tested additional uplift rate scenarios (Fig. S7), given that there has been concerns on the terrace chronology that we adopted (Brown and Bourlès, 2002), and other studies have suggested the terrace at 27 m elevation might be formed during MIS 5a, 5c or 5e instead of MIS 3 (Bradley and Addicott, 1968; Lajoie et al., 1975; Kennedy et al., 1982; Weber et al.,

1990). These uplift rates can fit the terrace sequence morphology equally well in terms of topographic misfit, but generally imply a larger possible range of paleo sea level. This can be explained by the increased terrace re-occupation for lower uplift rates (Malatesta et al., 2021), which also explains why the posterior ranges for the initial slope and wave base depth change increase for lower uplift rate scenarios (Fig. S7), and erosion rate estimates decrease. These tests suggest that locations with higher uplift rates will generally provide narrower constraints on paleo sea-level, while still providing realistic and unbiased parameter estimates.

# 5 Gulf of Corinth marine terrace sequence inversion

The complexity of semi-isolated basins, connected to sea during some time intervals and isolated in others, make them ultimate testing grounds for our modelling approach. We focus on one of such basins, the SE Gulf of Corinth, to derive a sea-/lake-level history from terrace sequence geometries, and compare its outcomes to paleoclimate data, tectonic structures and sill dynamics.

Natural interaction between the Gulf of Corinth and the open sea is currently restricted by the Rion and Acheloos-Cape Pappas sills at its W entrance at ~45-60 m depth (Fig. 4; Beckers et al., 2016). In the past there was an additional connection at its E end along the Corinth Isthmus, currently uplifted at ~80 m elevation but consisting of Quaternary marine sediments (Fig. 4; Caterina et al., 2023). These sills have controlled the Gulf's connection with the open sea over the past few hundred thousand years and lead to an alternation of marine and (semi-)isolated lake environments within the Gulf (McNeill et al., 2019). Although we approximately know the timing of these alternations, it remains unclear whether sill depths remained stable or fluctuated throughout the Quaternary (Roberts et al., 2009; McNeill et al., 2019), and in addition, whether lake levels were stable or fluctuating during periods with no connection to the sea. Hydroclimatic conditions, i.e. the balance between inflow, outflow, precipitation and evaporation, determine the level of the lake, and state-of-the-art paleo-environmental reconstructions (e.g. Kafetzidou et al., 2023) are insufficient to infer its fluctuations over time.

Terrace sequences are well exposed in the SE of the Gulf, where the Gulf of Corinth Fault System (Fig. 4) has lead to differential coastal uplift rates (Armijo et al., 1996; De Gelder et al., 2019; Fig. S8). This peculiarity allows us to test on a natural example whether the joint inversion of multiple terrace sequence profiles with different uplift rates provides a better-constrained sea-/lake-level history (as in Fig. 2). To account for the unknown range of possible lake-level elevations, we carried out inversions with all nodes from (semi-)isolated periods broadly constrained between -15 and -150 m elevation. For the marine periods we follow the more restricted eustatic sea-level ranges defined in Fig. 1bc (red boxes), as the resulting posterior sea-/lake-level ranges would remain similar to the prior ranges for tests in which we also gave a lot of freedom to nodes for the marine periods (Fig. S4). We selected three topographic profiles with little river incision and ~0.4-1.4 mm/yr uplift rates (Fig. S8), and avoided modelling the broad coastal plains at the base of all profiles that appear to have been modified by human presence (Fig. S8). We used the 90% percentile of 100-m wide swath profiles to obtain representative terrace sequence morphologies (Fig. S8). For the three profiles we assigned prior ranges of possible uplift rates of 1.25-1.4, 0.7-0.9 and 0.4-0.55 mm/yr (De Gelder et al., 2019; Fig. S8), and broad prior ranges for erosion rate (100-1500 mm/yr), initial slope (1-20%) and wave base depth (1-12 m).

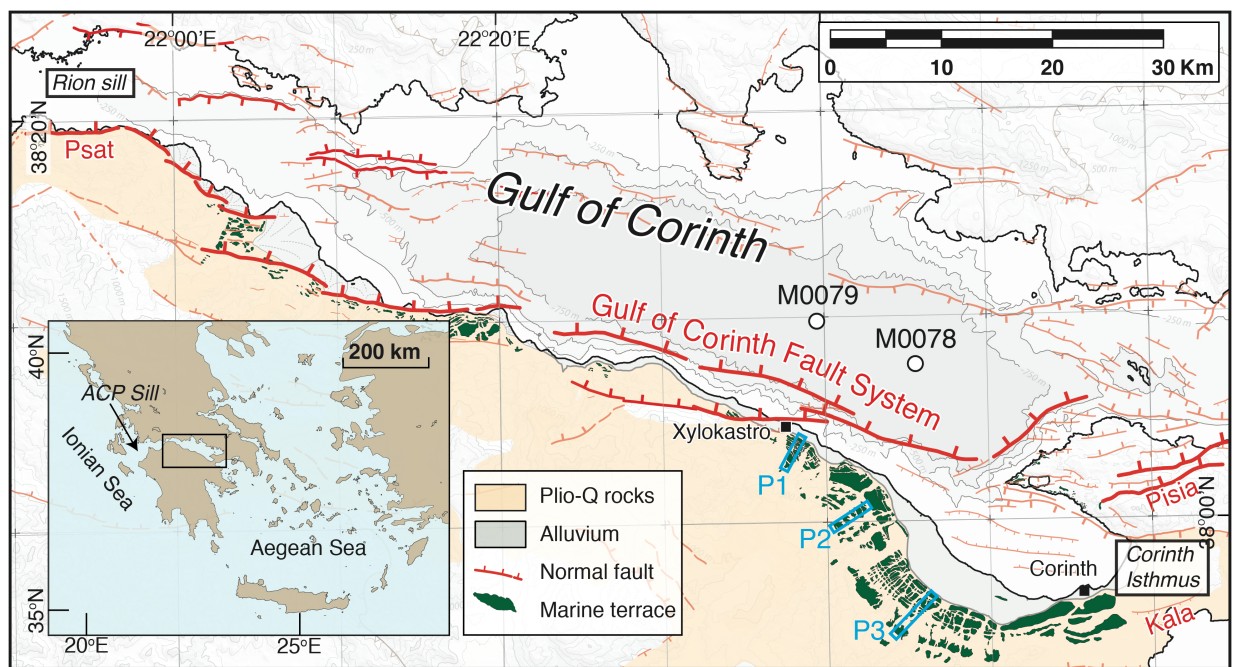

**Figure 4. Tectonomorphology of the Gulf of Corinth.** Map showing the main features of the Gulf of Corinth, including the active faults, marine terraces, profile locations used in the inversion and connections to the Ionian Sea (Rion Sill) and Aegean Sea (Corinth Isthmus). M0078 and M0079 indicate the IODP-381 sedimentary coring locations (McNeill et al., 2019). Psat = Psathopyrgos Fault, Pisia = Pisia Fault, Kala = Kalamaki Fault. Modified from de Gelder et al. (2019).

The individual profile inversions mostly constrain paleo sea/lake level for profile 1 (Fig. 5a), because it has the highest uplift rate and contains most terraces. The other two profiles provide limited constraints on paleo sea/lake level when inverted individually (Fig. 5b/c), but when jointly inverted with profile 1 they provide a much narrower range in terms of posterior distribution for sea-level (Fig. 5d). The cumulative misfit for the individual inversions (28 m) is slightly better than for the joint inversion (46 m), but there are no major visible differences between the terrace sequence profiles for the two inversions, and apart from the highest terrace of profile 2 (Fig. 5b) the terrace sequences are all near perfectly reconstructed. The three profiles show variations in initial slopes that are in line with the overall morphology, i.e. present-day profile 1 is steeper than 2, which is steeper than 3, which is also what we find for their initial slopes. The three profiles do have similar posterior distributions for wave base depths and erosion rates (Fig. 5e). Although we might expect lateral differences in these rates given variability in sediment types, catchment area and coastal orientation, the broad ranges for the posterior distributions indicate we cannot quantify these lateral differences from the profile morphology alone. The posterior parameter ranges mostly remain the same between the individual and joint inversion, with exception of the uplift rates for profiles 2 and 3 that became a little lower for the joint inversion. As for the sea-/lake-level inversion, all the other posterior parameter ranges become narrower for the joint inversion (Fig. 5e).

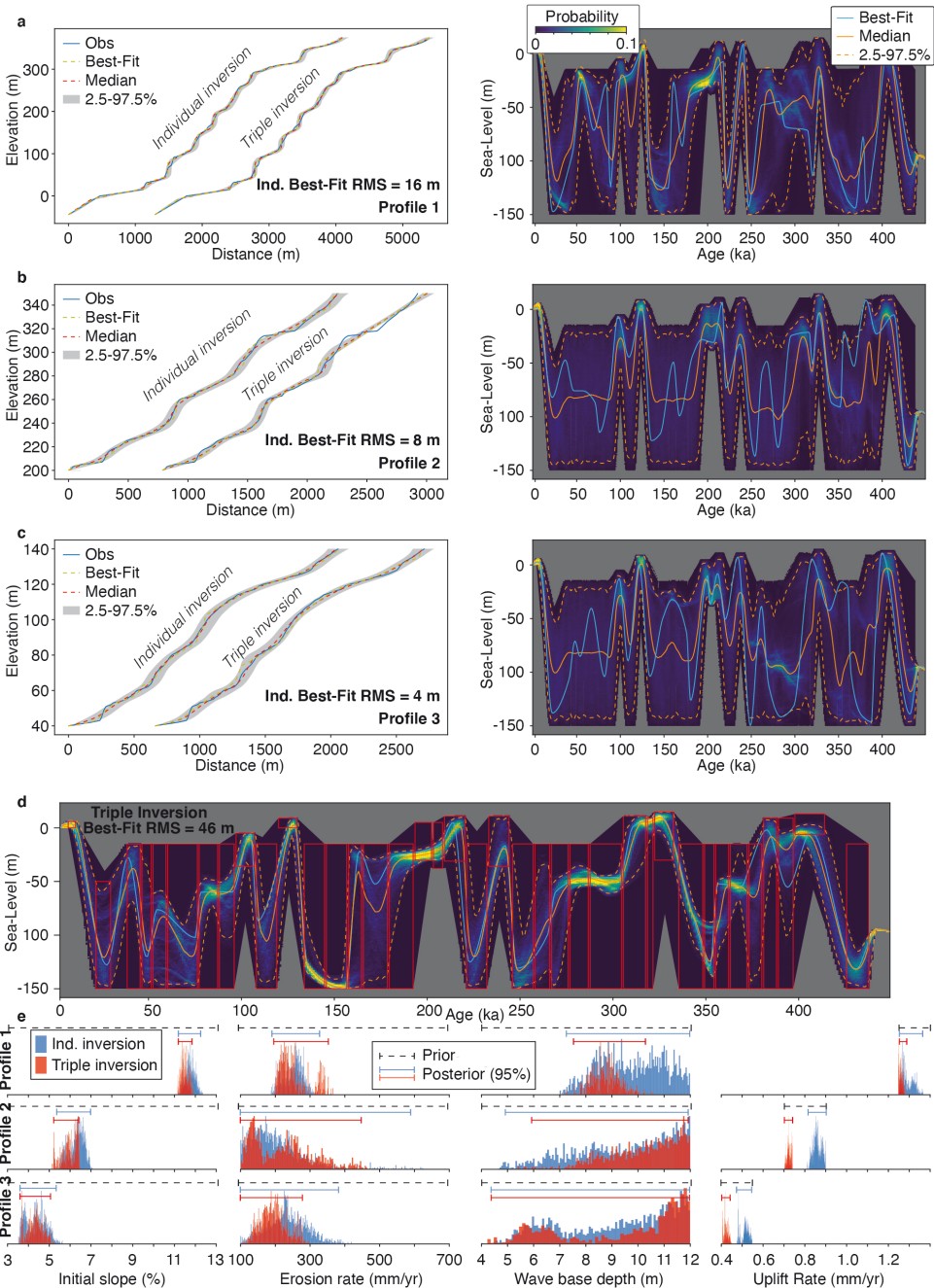

**Figure 5. Inversion of SE Corinth Rift marine terrace sequence (previous page). a-c)** Observed topography (left) from 3 different profiles in the SE Corinth Rift (locations see Fig. S8), together with the modeled best-fit, median, 2.5% and 97.5% percentile profiles for both an individual profile inversion and a joint inversion of the three profiles (horizontally offset by an arbitrary value). Corresponding probabilistic sea/lake-level ranges for the individually inverted profiles are given on the right, **d)** Posterior distribution for sea/lake-level history from joint inversion, with red boxes describing prior ranges **e)** Prior and posterior parameter ranges from both individual and joint profile inversions

The inverted sea-/lake-level history (Fig. 5d) shows a few notable features. To a first order, fluctuations resemble global sea-level trends, with relatively fast periods of sea-/lake-level rise prior to major sea-level highstands, followed by long periods of slow sea-/lake-level fall (Fig. 1c). Yet, unlike global sea-level trends, there are several periods of prolonged stability, in particular around 180-200 ka and 275-300 ka, and possibly also around 75-95 ka and 360-370 ka. In addition, glacial periods are often surprisingly poorly resolved, like during the period 20-75 ka or 160-180 ka. In the last section we discuss our interpretation of these trends, and show how they provide insightful arguments to decipher the relation between water level, fault activity, paleoclimate and tectonics.

## 6 Discussion

### 6.1 Beyond sea level: tectono-hydro-climatic processes in the Gulf of Corinth

While the results for the joint inversion of the 3 profiles in the Gulf of Corinth permit to unravel sea- and lake-levels through time as expected, we also stress that such inversions yield more, when integrated within the broader environmental setting: these reconstructions allow for a more detailed look into sea- and lake-level fluctuations within a (semi-)isolated basin. Figure 6 compares our inverted sea-lake level to the stratigraphy, facies and pollen content within two sedimentary cores from the sea floor of the central basin (McNeil et al., 2019; Gawthorpe et al., 2022; Kafetzidou et al., 2023). Based on those combined datasets, we propose that the main hydroclimatic modes that have occurred in the Gulf of Corinth throughout the past 450 ka, are 1) marine Gulf of Corinth, 2) transitional Gulf of Corinth or overfilled Lake Corinth and 3) underfilled Lake Corinth. The first 2 of those have been proposed before based on sedimentary cores (McNeill et al., 2019; Gawthorpe et al., 2022), whereas we base the occurrence of intervals with an underfilled Lake Corinth on our marine/lake terrace inversion, as available paleoclimatic proxies are unable to uncover them.

The major peaks in our reconstructed sea/lake-level curve occurred during interglacial sea-level highstands, as direct consequence of our choices in the ranges of the priors, when sea level in the Gulf of Corinth was similar to eustatic sea level (marine mode M; Fig. 7a). Sedimentary cores indicate marine conditions (McNeill et al., 2019), the corresponding stratal packages are bioturbated, and associated sedimentary facies are types FA1 and FA6 (Fig. 6; see caption for facies description). From pollen records, the typical reconstructed biomes are cool mixed evergreen needleleaf and deciduous broadleaf forests, indicating relatively warm and wet conditions with low amounts of steppic taxa (Kafetzidou et al., 2023; Fig. 6).

We interpret the interstadial periods around 75-95 ka, 180-200 ka, 275-300 ka and 360-370 ka as periods with an overfilled Lake Corinth, possibly with some marine incursions indicating a transitional Gulf of Corinth (T/O mode; Fig. 7b). This would explain the prolonged sea-/lake-level stability, during interstadial periods when eustatic sea-level fluctuated by tens of meters (e.g. Spratt and Lisiecki, 2016; De Gelder et al., 2022). In that case, sea-/lake-level elevations would correspond to the paleo-sill depth of the Rion Sill and/or Corinth Isthmus (white line, Fig. 6). Within the sedimentary cores, these periods are mostly characterized by laminated stratal packages, and associated sedimentary facies are types FA2, FA3 and FA4 (Fig. 6; see caption for facies description). The occurrence of marine incursions into Lake Corinth during these interstadial periods

is suggested by dated corals of ~76 ka, ~178 ka and ~201 ka (Roberts et al., 2009; Houghton et al., 2003) as well as the white, aragonite-rich laminations of FA3 and FA4. In other locations such laminations have been linked to (seasonal) mixing of marine and non-marine surface waters (Sondi and Juracic, 2010; Roeser et al., 2016).

The glacial periods are characterized by relatively low sea/lake-level elevations, possibly even down to the lower limit of -150 m we used in the inversion. We interpret these periods as underfilled Lake Corinth conditions (U mode; Fig. 7c), during which water inflow was lower than water evaporation within the lake, and lake level fell down to tens of meters below the sill depth. Sedimentary cores indicate non-marine conditions (McNeill et al., 2019), the corresponding stratal packages are mostly bedded and associated sedimentary facies are types FA5 and FA11 (Fig. 6; see caption for facies description). Reconstructed biomes from pollen suggest an increase in open vegetation such as grassland and steppe communities under colder and drier conditions (Kafetzidou et al., 2023; Fig. 6), matching reconstructed periods of lake underfilling. In general the inverted resolution of the lake-level elevation is much lower for these periods, with large probabilistic ranges. We attribute this to the occurrence of rapid lake-level fluctuations, like in other isolated E-Mediterranean water bodies such as the Dead Sea (Stein et al., 2010) and Lake Van (Turkey; Landmann et al., 1996). In such environments, the lake level is determined by the budget between runoff and evaporation, and quick variations are expected. Alternatively, this could also be due to the fact that terraces formed during low sea/lake level get increasingly eroded during transgressions.

The interstadial periods with prolonged sea-/lake-level stability also allow for a possible reconstruction of sill depths through time (white lines, Fig. 6). The westernmost sill, the Acheloos-Cape Pappas Sill (Fig. 4), is currently at a depth of ~45-48 m. While there are no major faults there, we can't exclude slow subsidence or uplift at a few tenths of mm/yr, cumulating to a few tens of meters on the 100 kyr time scale. The Rion Sill, at the western entrance to the Gulf of Corinth, is currently at ~62 m depth (Perissoratis et al., 2000). As it is located in the hanging wall of the Psathopyrgos Fault (Fig. 5), active since at least the past ~200 ka (Houghton et al., 2003), the Rion Sill was unlikely deeper in the past. We reconstruct the Rion Sill depth assuming marine incursions around ~76 ka (Roberts et al., 2009) took place through this sill, and the Rion Sill was not lower than sea/lake level during the overfilled/transitional interval around ~200 ka. Extrapolating the trend, it would make sense for the older connections between Lake/Gulf of Corinth and the open sea to have occurred primarily through the Corinth Isthmus at the eastern end of the Gulf of Corinth.

The Corinth Isthmus is currently at an elevation of ~80 m, and has been uplifted through the Pisia Fault, Kalamaki Fault and/ or a regional uplift (Armijo et al., 1996; Roberts et al., 2009; Caterina et al., 2022). We reconstruct the Corinth Isthmus depth assuming lake/sea levels during overfilled/transitional intervals around ~290 and ~360 ka correspond to the Corinth Isthmus depth, and the isthmus was not lower than sea/lake level during the overfilled/transitional interval around ~200 ka. Extrapolating this trend fits with the current Corinth Isthmus elevation of ~80 m. The isthmus elevation before ~360 ka is difficult to constrain from our data, but was possibly shallower before, given the small amount of marine sediments deposited around the ~400 ka interglacial period, and the lack of deposits within the Isthmus stratigraphy older than ~350 ka (Collier and Dart, 1991; Caterina et al., 2022). Our reconstruction of both Rion Sill and Corinth Isthmus fits with the sedimentary interpretation of a tidal strait around ~300 ka at the isthmus, with marine connections on both ends of the Gulf of Corinth (Caterina et al., 2022).

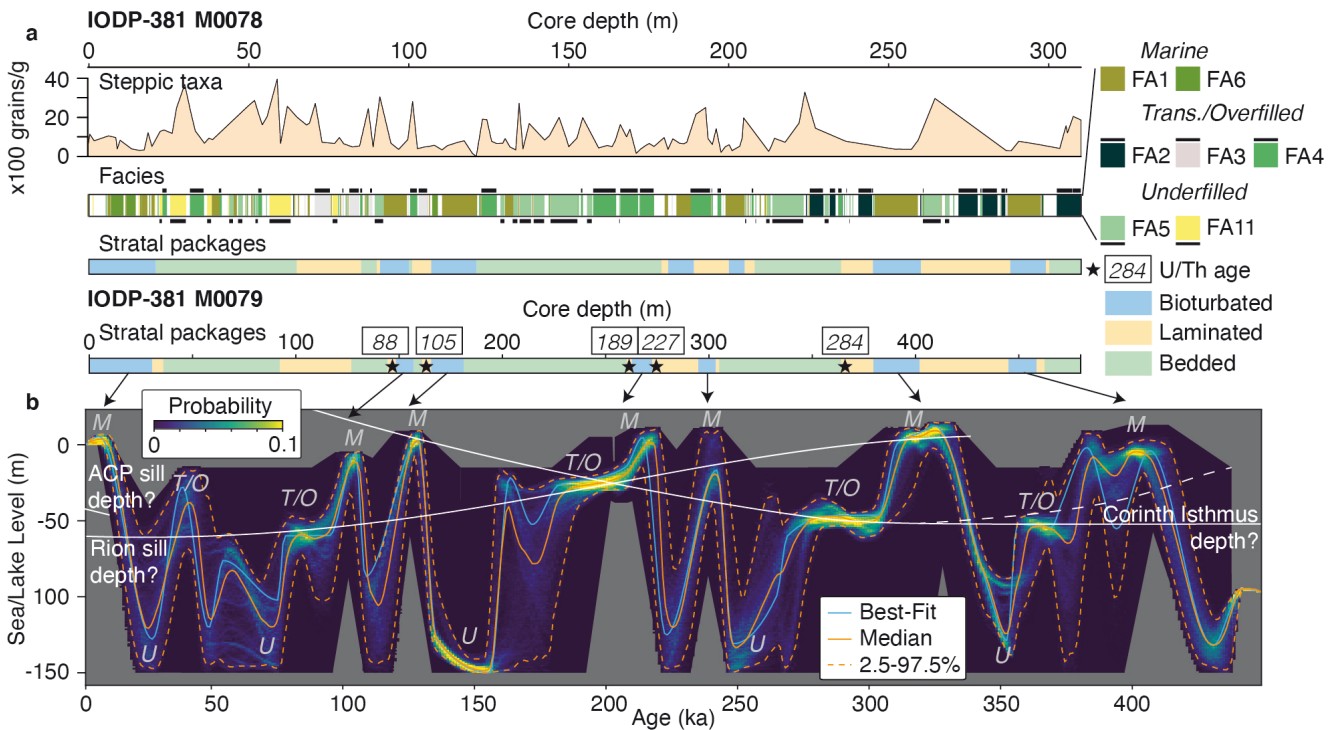

**Figure 6. Comparison of sea-/lake-level reconstruction to other datasets. a)** Comparison to IODP-381 cores M0078 and M0079, with steppic taxa from Kafetzidou et al. (2023), facies from McNeill et al. (2019) and stratal packages with U/Th ages from Gawthorpe et al. (2022). FA1: homogenous mud, FA2: greenish gray mud with dark gray to black silty-to-sandy beds (cm-scale), FA3: light gray to white sub-mm laminations (cc or aragonite) alternating with mud–silt beds, FA4: laminated greenish gray to gray mud with muddy beds, FA5: greenish gray mud with homogeneous cm thick gray mud beds, FA6: green bedded partly bioturbated mud, silt and sand, FA11: interbedded mud/silt and cm thick sand beds. **b)** Inversion result from Fig. 5d with proposed sill/isthmus elevations, marking periods with marine (M), transitional or overfilled (T/O) and underfilled (U) hydroclimatic modes.

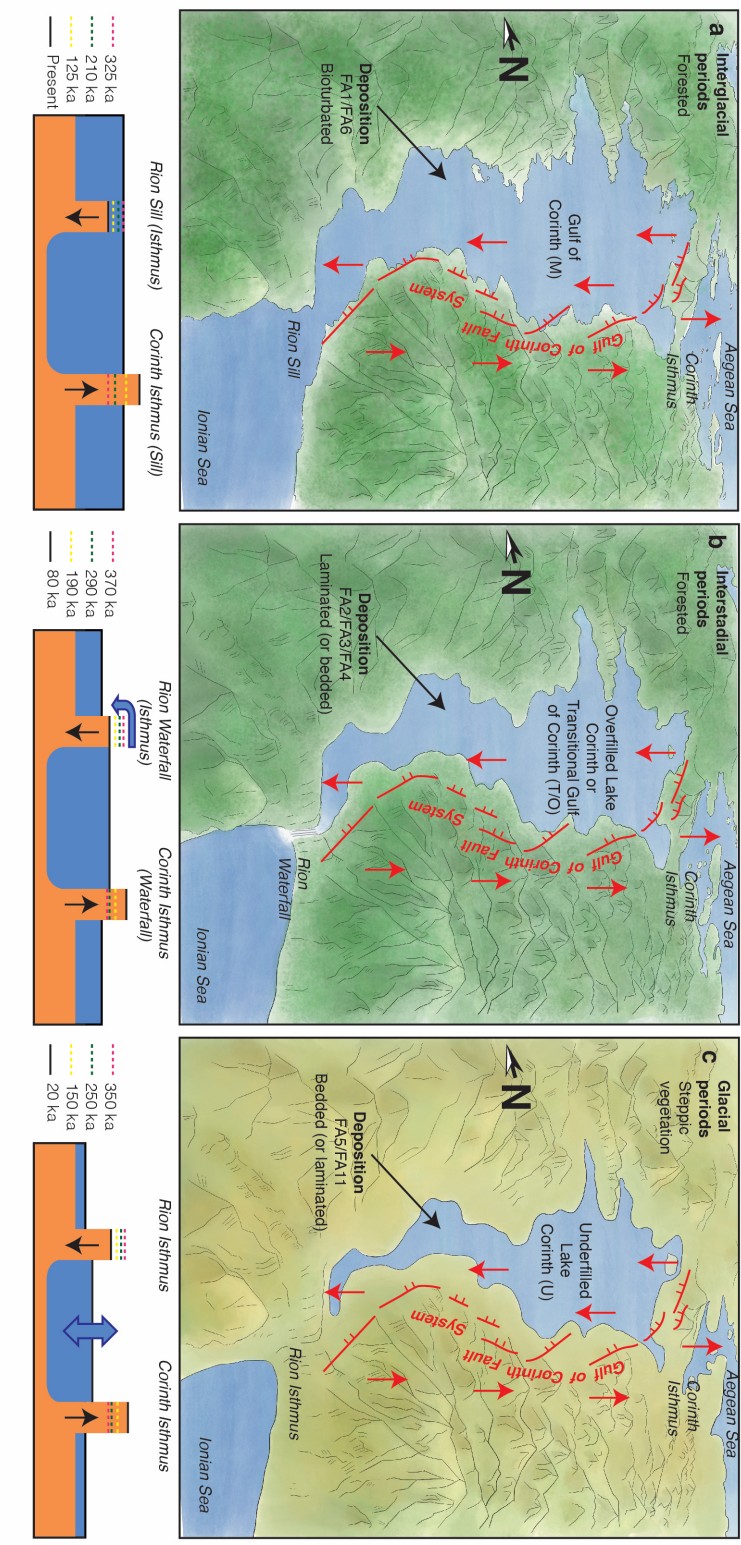

**Figure 7 (Previous page). Schematic illustrations of the different hydroclimatic modes in the Lake/Gulf of Corinth.** Illustrations of **a)** the interglacial marine (M) hydroclimatic mode as it is today. Around MIS 5e (~125 ka) the marine connection also occurred above the Rion Sill, with the Corinth Isthmus a land bridge like it is today. Around MIS 7a-e (~210-240 ka) and MIS 9 (~325 ka) the marine connection occurred above both the Rion and Corinth Sills. Around MIS 11 (~400 ka) the marine connection would have occurred above the Corinth Sill, with a Rion Isthmus on the W end of the Gulf of Corinth. **b)** The transitional or overfilled (T/O) hydroclimatic mode at ~80 ka, with a narrow connection or waterfall to the open sea at the Rion Sill. Around ~190 ka a narrow connection or waterfall to the open sea could have occurred at both the Rion and/or Corinth Sill, and at ~290 ka and ~370 ka only at the Corinth Sill, with a Rion Isthmus on the W end of the Gulf of Corinth. **c)** The underfilled (U) hydroclimatic mode at ~20 ka. As for the older glacial periods (~150, ~250 and ~350 ka), land bridges occur on both side of Lake Corinth, at the Rion and Corinth Isthmus. In all illustrations red arrows indicate uplift and subsidence in the footwall and hanging wall of the Gulf of Corinth Fault System, respectively.

Our exploration of sea-/lake-level variations in the Gulf of Corinth demonstrates the strength of using marine/lacustrine terrace sequence inversion. Although several questions remain – like the effects of erosion and sedimentation on sill evolution, or the effects of non-constant uplift rates of the marine terrace sequences – we are able to provide a solid framework that can explain several different tectonic and hydro-climatic processes simultaneously. Furthermore, our results may provide a window to explore the links between paleogeographic and biologic evolution, with our proposed history of sills, waterfalls and land bridges (Fig. 7) theoretically impacting temporal isolation and mixing of species between the Peloponnese and the Greek mainland. We distinguish 4 different hydroclimatic modes of for the Lake/Gulf of Corinth, that have probably also occurred in other (semi-)isolated basins like the Sea of Marmara or the Carioco Basin. Marine and transitional modes will most likely depend on eustatic sea-level elevations and sill depths, whereas over- or underfilled lakes likely depend on sill depths and local climatic conditions. For the Corinth Lake we show that this transition from over- to underfilled lakes occurs during changes from interstadial to glacial periods, and is accompanied by changes in vegetation that imply drier conditions.

## 6.2 Inversion of marine terrace sequences

In the examples above, we showed how to assess paleo sea-level variations, and simultaneously extract quantified metrics for morphotectonics and hydrodynamics, from the geometry of marine terrace sequences. Using a probabilistic inversion methodology set in a Bayesian framework, we avoid the simplifications of bijective approaches in which a single marine terrace is always linked to a single sea-level highstand and vice-versa (e.g. Pastier et al., 2019; Malatesta et al., 2021). By considering a full sea-level curve and its possible variability, it is possible to provide quantitative constraints on highstands, lowstands, sea-level rise and fall, filling the observational gap for time periods for which field measurements are scarce. We admit that some model simplifications and approximations may alter our interpretations. In particular, we neglect subaerial erosion, and kept uplift rate, erosion rate, initial slope and wave base depth parameters time-constant for each individual sampled paleo sea level curve. Both could be fine-tuned in future developments.

To apply this model procedure to other sites, we can make several recommendations based on our findings in this paper. Any coastal area with marine terraces will have a lateral variability in terms of terrace width and terrace elevation. Given that the posterior ranges for the model parameters (paleo sea-level, uplift rate, etc.) will directly depend on the chosen inversion parameters (*ipstep*, *σ* and *corrl*), it is important to adjust those inversion parameters to the naturally observed variability in the geometry of a marine terrace sequence. In the Santa Cruz example the similarity in terrace width/height variability between the modeled posterior ranges (Fig. S5), and the observed variability within a section of the marine terrace sequence

(Fig. S6) justifies our choice of inversion parameters. But, if we wanted to characterize a larger area along the Santa Cruz coastline, we would either have to increase $\sigma$ and/or **corrl** to account for the higher degree of variability, or jointly invert multiple cross-sections in which every cross-section has a similar variability in terrace height/width.

The prior information required to obtain reliable results, will also be dependent on the context of a specific marine terrace sequence, and on the parameter cut-off choices deemed realistic. For Santa Cruz and Corinth, the posterior distributions for wave base depth suggest that, purely based on marine terrace sequence morphology, this cut-off could have been deeper than 10-12 m (Figs. 3 and 5). However, based on observations and models of cliff erosion it seems unlikely that the wave base for bedrock erosion is more than 10 m for Santa Cruz (Kline et al., 2014), and wave base depths/heights should be smaller for the calmer Gulf of Corinth. This choice in cut-off in turn affects the posterior distribution of other parameters, and should thus be chosen carefully.

In the cases of Corinth and Santa Cruz we assigned relatively narrow windows for the uplift rate, based on chronological information that was already available. As such we only obtained refinements, rather than 'new' information about the uplift rate. For the Santa Cruz case, we tested four uplift rate scenarios matching different possible chrono-stratigraphies (Fig. S7) that could all explain the morphology equally well, albeit with different ranges for the other parameters. We also inverted the Santa Cruz terrace sequence morphology with a broad range between 0.3 and 1.5 mm/yr, but the inversion would converge on only one of the four scenarios in Fig. S7. This suggests that at least an approximate prior idea on the uplift rate is a prerequisite for a reliable inversion of a marine terrace sequence morphology.

Many paleo sea-level studies that use geomorphic/geologic observations tend to have a confirmation bias regarding sea-level curves, and propose refinements of paleo sea-level estimates to sub-m scale (e.g. Murray-Wallace, 2002; Roberts et al., 2012) or uplift rates to precisions of ~0.01 mm/yr (e.g. Pedoja et al., 2018; Meschis et al., 2022). In this study, we take a step back by allowing more freedom to possible paleo sea-level variations, as well as uplift rate, erosion rate, initial slope and wave base depth, to provide a more reliable way to translate morphologic observations to paleo sea-level constraints. For instance, the low uplift rate examples from the Corinth Rift (Fig. 5b, c) and Santa Cruz (Fig. S7) reveal very little about paleo sea/lake levels, even if the uplift rate is roughly known. As a marine terrace is formed over several sea-level cycles, the resulting terrace width and height will depend on all those cycles, as well as wave base depths, erosion rates and initial slopes, all of which are generally poorly constrained. Even in the hypothetical case that these parameters are known (Fig. 2), there is still a wide spectrum of sea-level histories that could have created the specific morphology of a marine terrace sequence. It suggests that estimating paleo sea-level based on the comparison of a present-day landform to a paleo-landform (Rovere et al., 2016), may be too simplistic in many cases, at least for erosive marine terraces. Although uncertainties that we provide on paleo sea-level are much larger than what calculations based on hydrodynamic ranges would suggest (Lorchsteid and Rovere, 2019), we do consider them to be reliable as they take in a large number of unknowns. To reduce uncertainties, we demonstrated in this paper that a joint inversion of multiple profiles is a powerful way forward.

For simplicity, as the aim of this paper was to test the model on increasingly complex settings, we only inverted 1 profile for Santa Cruz and 3 for Corinth, but it is easily possible to explore with more profiles at those locations in future work. In addition, although here we focused on erosive marine terraces to develop a proof of concept, another promising avenue is to apply this inversion method to bio-constructed (coral reef) terraces, which tend to be better dated (e.g. Pedoja et al., 2014;

Hibbert et al., 2016) and for which modelling routines also exist (e.g. Toomey et al., 2013; Pastier et al., 2019). One of our key findings is that inverting multiple profiles simultaneously provides much better paleo sea-level constraints than focusing on individual profiles (Figs. 2, 5). The global archive of paleo-shorelines (Fig. 1a) presents a huge potential for such multi-profile marine terrace inversions. This massive inversion would not only lead to improved estimates of local relative sea-level histories, but may also complement studies on glacio-isostatic adjustments that are relevant to a global sea-level perspective.

## 7 Conclusions

In this study we demonstrated the use of a probabilistic inversion approach to decipher the formation of marine terrace sequences in general, and the tectonic and hydro-climatic evolution of (semi-)isolated basins in particular. With this approach, we provide the tools (see below) to simultaneously estimate past sea-level variations, uplift rates, erosion rates, initial slopes and wave heights.

From synthetic tests, benchmarking on a terrace sequence near Santa Cruz marine terrace sequence, and application to the sequence in the SE Gulf of Corinth, our results bring a theoretical advance by showing that : 1) Paleo sea-level and other parameter ranges can be better constrained from sequences that are uplifting at higher rates compared to lower rates, and better constrained from a joint inversion of multiple profiles than from inversion of a single profile. 2) Uplift rates, sea-level variations and wave erosion parameters are intricately linked. By allowing more freedom to possible ranges of all the relevant parameters, we provide a more reliable way to translate morphologic observations to paleo sea-level constraints. Resulting uncertainties may be higher compared to 'classic' approaches of comparing present to past shoreline elevations, but are more realistic. 3) Probabilistic inversion of marine terrace sequences is a powerful method, applicable to a large portion of the world's coastlines to disentangle tectonic and hydro-climatic processes.

Beyond the methodological achievement, by applying our method to a complex case -the semi-isolated Gulf of Corinth (Greece), we show that this method can be a powerful tool to explore subtle environmental forcings, like the balance between precipitation and evaporation, that may have had a prime importance in setting the lake level during certain periods of time. We found that eustatic sea-level and tectonically changing sill depths drive marine and transitional phases during interglacial and interstadial periods, respectively. Wetter and drier conditions drive over- and underfilling of Lake Corinth during interstadial and glacial periods, respectively. We expect such transitions to be different for each unique tectono-hydro-climatic setting, with our inversion approach providing a new way to decipher such geomorphic Rosetta stones.

## Code availability

The marine terrace inversion code used in this study can be found at https://github.com/ginodegelder/Rosetta.

## Supplement link

The supplementary material to this study can be found at https://doi.org/10.31223/X5B117.

## Author contribution

**Conceptualization** by GdG, LH and TB, **Formal analysis** by GdG, **Funding acquisition** by TB, **Investigation** by GdG, NH, LH and TB, **Methodology** by GdG, NH, AMP, TB and YB, **Visualization** by GdG, NH and YB, **Writing - original draft** by GdG, **Writing - review & editing** by all authors.

## Competing interests

The authors declare that they have no conflict of interest.

## Acknowledgements

This study has been funded by the European Union horizon 2020 research and innovation program under grant agreement 716542. Furthermore, GdG acknowledges postdoctoral funding from the IRD and the Manajemen Talenta BRIN fellowship program, and research permit 52/SIP.EXT/IV/FR/5/2023 provided by the Indonesian government on May 11th 2023 . GdG likes to thank Katerina Kouli for sharing her pollen data, and other IODP-381 expedition members, as well as David Fernández-Blanco, Robin Lacassin and Rolando Armijo, for the many fruitful discussions on the Corinth Rift. We thank Dilruba Erkan for the illustrations of the Corinthian landscapes.

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
