# Peer review of "Bayesian reconstruction of sea-level and hydroclimates from coastal landform inversion: application to Santa Cruz (US) and Gulf of Corinth"

_EGUsphere, 2024_

## Author Comment (AC1)

Dear Reviewer,

We provide hereunder an overview of how we address the comments, with explanations and references, on a point-by-point basis. Reviewers' comments are in black and our response in blue. In the manuscript, all the new/changed text is highlighted in yellow.

**Reviewer 1 Comments to Author:**

De Gelder et al., present a novel inversion approach to estimate marine terrace formation parameters and sea level histories from marine terrace topographic data. They use a set of test scenarios to demonstrate the feasibility of the approach and apply it to well-studied marine terrace sequences in Santa Cruz and the Gulf of Corinth. The topic and scope of the study are very promising. Using inverse schemes to study marine terrace sequences is a logical next step for the community and will be of interest to many people. This study therefore holds a lot of promise and comes with nice figures, however, the authors omitted a lot of critical detail that is necessary when describing a new method.

Thank you, we appreciate the comment. For more details on the method, see comments below.

Unfortunately, the authors uploaded a preprint without line numbers, therefore, I added some more descriptive text locations.

Our apologies, we forgot to add these.

The authors measure the horizontal misfit with topography and therefore focus on the width of terraces. This is interesting because terrace width is probably an underused terrace parameter that holds information. At the same time, terrace width (and topography I general) is often highly variable along-strike. It would be interesting to include a real case, where uplift rate is roughly constant, and several topographic profiles of the same sequence are jointly inverted. This would show whether this approach is robust enough to cope with along-strike topographic variability.

Actually the horizontal misfit is measured at every vertical step (istep), and so the misfit takes into account both the horizontal and vertical variability of the terrace sequence. We gave more details about the misfit calculation (Line 146-158):

*"In this work, the data vector is defined as a set of points measured on the shoreline with a vertical step size (ipstep; Fig. 2a). The misfit between this observed topography and the modelled paleo-shoreline sequences is calculated on the horizontal axis. The elevation axis is therefore divided in a regular grid, and the level of data fit is measured by comparing the horizontal distance between observed and simulated horizontal position at each grid point. In this way, the terrace width is used as an observed parameter. Uncertainties about the observed shorelines account for the inability of our numerical model to explain observations, and can be due to both observational or modelling errors. These errors are supposed Gaussian and described by a standard deviation (σ; Fig. 2a) and the level of spatial correlation related to the spatial resolution of the data (corrl; Fig 2a). In preliminary tests (Fig. S2) we also tested inversions with the misfit calculated along vertical axes (i.e. comparing elevations between observed and estimated profiles), but found it harder to reproduce realistic marine terrace sequences with a few m of terrace height variability, and a few hundred meter of terrace width variability (e.g. Regard et al., 2017; De Gelder et al., 2020). We note that changing the misfit calculation from horizontal to vertical misfit does not change the paleo sea-level posterior distribution (Fig. S2)."*

To go further into horizontal and vertical uncertainties, we added 1) the horizontal and vertical ranges for the 2.5-97.5 percentiles visually in Fig. S4, as well as the average horizontal and vertical ranges for the different parameter variations, 2) an extra figure about the Santa Cruz terraces, showing how the chosen ranges approximately match the horizontal and vertical uncertainties in a real case, and 3) additional discussion about horizontal and vertical uncertainties in the text, also referring to this new supplementary figure (lines 440-449):

*"To apply this model procedure to other sites, we can make several recommendations based on our findings in this paper. Any coastal area with marine terraces will have lateral variability*

*in terms of terrace width and terrace elevation. Given that the posterior ranges for the model parameters (paleo sea-level, uplift rate, etc.) will directly depend on the chosen inversion parameters (ipstep, σ and corrl), it is important to adjust those inversion parameters to the naturally observed variability in the geometry of a marine terrace sequence. In the Santa Cruz example the similarity in terrace width/height variability between the modeled posterior ranges (Fig. S4), and the observed variability within a section of the marine terrace sequence (Fig. S5) justifies our choice of inversion parameters. But, if we wanted to characterize a larger area along the Santa Cruz coastline, we would either have to increase σ and/or corrl to account for the higher degree of variability, or jointly invert multiple cross-sections in which every cross-section has a similar variability in terrace height/width."*

The authors should include the equation of the forward model to make the methods easier to follow.

We added some text and the main equation to make the details of the forward model more clear (Lines 109-116):

*"The initial erosion rate ER is expressed as an effective eroded volume per unit of time and coastal length, in which a fraction of erosional residual power ERr erodes the foundation at each location along the profile. This fraction ERr depends on the water level, h, the water depth for wave base erosion WB, and a coefficient for sea bed erodibility K, so that:*

*Then a residual power (ER - dERr /dt) carves out a 1 m high notch and all its overhanging material to form a cliff. Following previous studies (Anderson et al., 1999; Pastier et al., 2019), for K we use 0.1 as bedrock erodibility and 1 for notch carving."*

A supplementary table summarizing all the paleo-sea level ranges, including a justification, should be added to the supplement.

Good suggestion, we added such a supplementary table (Table S1), and in the supplementary text we added the sentences "For MIS 10b and 10c we used the minimum from Spratt & Lisiecki (2016) and the maximum from the MIS 10a box." (which we had previously forgotten to mention) and "We summarize the ranges and justifications in Table S1 on the following page."

The methods require significantly more detail. Currently, the authors do not provide an adequate description of the algorithm. For instance, there is no mentioning of the type of sampler used in the MCMC. How many individual random walks are performed per scenario? Currently, only the number of forward simulations is mentioned. What acceptance ratio was aimed for? How was burn-in defined? What about autocorrelation? Effective sample size? ETC. This paper describes an exciting new inverse algorithm for marine terraces, and as such is lacking a lot of critical information. Citing an inverse problem review paper is not sufficient. This lack of detail also makes it harder to assess the presented study. None of the standard diagnostic plots of an MCMC inversion are presented, and it is therefore, hard to assess the performance of the algorithm.

More details have been given to describe the Bayesian inversion procedure and the McMC algorithm (Lines 125-141):

*"Reconstructing the sea-level history from present day observations of marine terrace sequences can be mathematically formulated as a highly non-linear inverse problem where the solution is non-unique. To embrace this non-uniqueness, the problem can be cast in a Bayesian (probabilistic) framework where the solution is a posterior probability distribution describing the probability of the model parameters (here the past sea-level variations), given the observed data (here the geometry of marine terraces). We use a Markov chain Monte Carlo algorithm to sample the posterior distribution and explore the range of models that can explain the observed topography within errors. In the Monte Carlo exploration of the model space, nodes can either be fixed at certain ages and elevations, or left free to move within a prescribed range (e.g. red boxes in Fig. 2b,d,e). The 4 main erosion model parameters (IS, ER, WB, UR) can also be fixed to chosen values, as done in the synthetic tests below, or left free within chosen ranges, as done for the Santa Cruz and Corinth examples below. The algorithm samples the parameter space as a random walk, where at each iteration a new sea level history model is proposed by perturbing the current one. The proposed model is then*

*either accepted or rejected following an acceptance rule based on the level of data fit of the current and proposed models. The final solution is a large ensemble of paleo sea-level models that approximates the probabilistic solution. That is, the distribution of models follows the posterior probability solution. Of the 1 million model runs per figure, the first 50 000 models were discarded as burn-in models. To verify that the random walk samples the target distribution, we show a number of convergence diagnostics in Fig. S1, which includes parameter acceptance ratios and likelihood evolution for all model runs in this article.“*

In order to show that the MCMC sampler converged towards the posterior distribution, we added in supplements some figures showing standard convergence diagnostics: Supplementary Figure (Fig. S1) was added, with plots of parameter acceptance ratios and log likelihood (misfit) values in relation to the model runs for all the main figures of the paper.

Priors: The authors describe the model parameters, but do not specify how these are treated in the inversion. The authors refer to a prescribed range, and therefore, I assume, use flat priors. But this should be specified. Also, it would help to stick to common terms in Bayesian frameworks, such as priors.

Yes, we treat them as flat priors and have specified as such now (Lines 158-160):

*“The assigned ranges for input parameters are all treated as flat uniform priors, i.e. a prior distribution that assigns equal probability to all the values within the specified ranges.”*

The authors measure the difference between observed and modelled topography in the horizontal axis, putting much emphasis on the width of terraces. How do the presented results change for measuring the vertical misfit? Commonly, terrace studies focus on terrace elevations and not their width.

Although the misfit is calculated on the horizontal axis, this does not imply there is no importance attributed to terrace elevations. We added Figure S2, and also specified in the text (Lines 154-158):

*“In preliminary tests (Fig. S2) we also tested inversions with the misfit calculated along vertical axes (i.e. comparing elevations between observed and estimated profiles), but found it harder to reproduce realistic marine terrace sequences with a few m of terrace height variability, and a few hundred meter of terrace width variability (e.g. Regard et al., 2017; De Gelder et al., 2020). We note that changing the misfit calculation from horizontal to vertical misfit does not change the paleo sea-level posterior distribution (Fig. S2).”*

Why are so many model parameters fixed in the test scenarios? Would be interesting to explore additional scenarios, to see which information is required for a given terrace sequence and when the inversion is not able to recover the parameters. Another important question is, how the inversion performs when the problem gets under-constrained. Does the algorithm converge on a (fake-) solution in a local minimum or does the MCMC roam the parameter space as it should.

We fixed most of the model parameters in the test scenarios to keep things simple initially, and increase the complexity later in the paper. Nevertheless, it is an interesting suggestion, so we added a similar test to the one in Figure 2, but with larger ranges on the priors for initial slope, uplift rate, erosion rate and wave base height (see Figure S3). This allowed us to verify that the inversion performs well and is able to sample a wide posterior distribution without getting trapped in local minima. We added some lines on this figure in the main text as well (Lines 212-217):

*“To understand what happens in scenarios in which more parameters are unknown, we repeated the same tests (from Figures 2b-d) with broad prior ranges for the uplift rate, initial slope, erosion rate and wave base height (Fig. S3). Also in this case the joint inversion provides much narrower posterior ranges for paleo sea-level compared to the two individual profile inversions, and not too different from the cases with fixed model parameters (Figs. 2b-d). The posterior ranges for all parameters are consistently smaller for the joint inversions, compared to the individual inversions (Fig. S3).”*

The authors do not define how they report inversion results. E.g., for the Santa Cruz inversion, they state the posterior range of parameters but do not explain, whether these are confidence intervals, a standard deviation around the mean, or similar. Also, the authors state that the model limits the uplift rate to 1.35-1.65 mm/yr, but this is the prior range and was therefore set beforehand. The word "limits" could also imply that the authors are in fact referring to the prior range. However, posterior range results are following, such as the range for initial slope. I know I am being nitpicky with this sentence, but here and elsewhere, imprecise language concerning aspects of the inversion creates confusion.

We appreciate the comment, and adjusted the text at several points for clarification (Lines 164, 237, 257, 258, 261, 262, 270, 277-279, 306, 307, 318, 326, 341, 342). In addition, we clearly differentiate the prior and posterior ranges now in Figures 3 and 5.

There seems to be a degree of circularity in the approach. Narrow uplift rate prior ranges are defined for the Corinth and Santa Cruz models, based on the elevation of dated marine terraces. These prior uplift rates ranges are then used to reproduce the stair-case morphology and invert for uplift rate, which was already an implicit input. Is this OK because the focus of the study is on the width of terraces and resolving parameters other than uplift? The authors should address this.

We added a small paragraph on the uplift rate in the discussion now (Lines 457-464):

*"In the cases of Corinth and Santa Cruz we assigned relatively narrow windows for the uplift rate, based on chronological information that was already available. As such we only obtained refinements, rather than 'new' information about the uplift rate. For the Santa Cruz case, we tested four uplift rate scenarios matching different possible chrono-stratigraphies (Fig. S6) that could all explain the morphology equally well, albeit with different ranges for the other parameters. We also inverted the Santa Cruz terrace sequence morphology with a broad range between 0.3 and 1.5 mm/yr, but the inversion would converge on only one of the four scenarios in Fig. S6. This suggests that at least an approximate prior idea on the uplift rate is a prerequisite for a reliable inversion of a marine terrace sequence morphology."*

We agree that concerning the uplift rates, the inversion mainly re-fines previous estimates, which we feel is appropriate for the current study, given that it focuses on sea level variations. Of course, alternative explorations on uplift rates -or even time varying uplift rates- could have been carried out, but that was beyond our scope here.

For the Corinth case, the posterior distributions of wave base depth in profile 2 & 3 have their maximum values at the boundary of the prior range, suggesting the algorithm would like to go to even deeper wave base depths. The authors, do not mention the results for the IS, ER, WB, UR parameters. These results should also be described and the implications of the posterior distribution ramping up against the prior boundary should be discussed, since this may be a problem.

We now expanded the description of IS, ER and WB parameters in Section 5 now (Lines 321-326):

*"The three profiles show variations in initial slopes that are in line with the overall morphology, i.e. present-day profile 1 is steeper than 2, which is steeper than 3, which is also what we find for their initial slopes. The three profiles do have similar posterior distributions for wave base depths and erosion rates (Fig. 5e). Although we might expect lateral differences in these rates given variability in sediment types, catchment area and coastal orientation, the broad ranges for the posterior distributions indicate we cannot quantify these lateral differences from the profile morphology alone. "*

… and also added more discussion about the posterior distributions ramping up against the limits we set (Lines 450-456):

*"The prior information required to obtain reliable results, will also be dependent on the context of a specific marine terrace sequence, and on the parameter cut-off choices deemed realistic. For Santa Cruz and Corinth, the posterior distributions for wave base depth suggest that, purely based on marine terrace sequence morphology, this cut-off could have been deeper than 10-12 m (Figs. 3 and 5). However, based on observations and models of cliff erosion it*

*seems unlikely that the wave base for bedrock erosion is more than 10 m for Santa Cruz (Kline et al., 2014), and wave base depths/heights should be smaller for the calmer Gulf of Corinth. This choice in cut-off in turn affects the posterior distribution of other parameters, and should thus be chosen carefully."*

Currently, the discussion ends with a lengthy paragraph of the Gulf of Corinth case. As a reader, this is a bit weird. Until here, the focus of this paper was the inversion method. However, the long Corinth section hangs at the end like an afternote. To improve flow and readability, I'd suggest to condense this section.

We feel like the implications of the Corinth inversion are very interesting, and merit a deeper discussion beyond the methodological advances that this paper presents. We think some readers will appreciate such a discussion, and have thus decided to keep this section as detailed as it is. We did change the order of sections 6.1 and 6.2, as to end the discussion in a broader context.

To emphasize more that we find the discussion of Corinth a key part of the paper, we modified the title, which is now;

*"Bayesian reconstruction of sea-level and hydroclimates from coastal landform inversion: application to Santa Cruz (US) and Gulf of Corinth"*

and added several sentences throughout the paper to help that emphasis (Lines 76-81, 225-229, 293-296, 346-349, 355-356, 503-506).

The authors present an exciting new tool and there are many things that could be discussed, but currently are not. What about typical lateral variability of coastline morphology and its influence on inversion results? What prior knowledge is typically needed to recover reliable results? What parameter trade-offs typically exist? Etc.

We've added some discussion about lateral variability (Lines 440-449):

*"To apply this model procedure to other sites, we can make several recommendations based on our findings in this paper. Any coastal area with marine terraces will have lateral variability in terms of terrace width and terrace elevation. Given that the posterior ranges for the model parameters (paleo sea-level, uplift rate, etc.) will directly depend on the chosen inversion parameters (ipstep, σ and corrl), it is important to adjust those inversion parameters to the naturally observed variability in the geometry of a marine terrace sequence. In the Santa Cruz example the similarity in terrace width/height variability between the modeled posterior ranges (Fig. S4), and the observed variability within a section of the marine terrace sequence (Fig. S5) justifies our choice of inversion parameters. But, if we wanted to characterize a larger area along the Santa Cruz coastline, we would either have to increase σ and/or corrl to account for the higher degree of variability, or jointly invert multiple cross-sections in which every cross-section has a similar variability in terrace height/width."*

… and prior knowledge (Lines 450-456):

*"The prior information required to obtain reliable results, will also be dependent on the context of a specific marine terrace sequence, and on the parameter cut-off choices deemed realistic. For Santa Cruz and Corinth, the posterior distributions for wave base depth suggest that, purely based on marine terrace sequence morphology, this cut-off could have been deeper than 10-12 m (Figs. 3 and 5). However, based on observations and models of cliff erosion it seems unlikely that the wave base for bedrock erosion is more than 10 m for Santa Cruz (Kline et al., 2014), and wave base depths/heights should be smaller for the calmer Gulf of Corinth. This choice in cut-off in turn affects the posterior distribution of other parameters, and should thus be chosen carefully."*

We already mentioned trade-offs between timing and elevation of sea-level peaks (Lines 197-200, 251-253), and correlation between erosion rate and wave base depth (Lines 254-256) and did not feel we needed to go further into parameter trade-offs; in a way, this study can be regarded as a proof of concept, also designed to release the code and develop technical aspects, before it can be used for broader parametric studies and regional applications, by anyone interested.

Section 2, second paragraph: There seems to be a typo in the reference (REEF).

It was not a typo, but the name of the code that we used. We clarified that now (Line 104):

*"The landscape evolution model we use (the code 'REEF'; Husson et al., 2018; Pastier et al., 2019) "*

We very much appreciate the time and effort by the reviewer to go through our manuscript, as well as the good suggestions made to improve the manuscript.

We hope the applied changes will be appreciated by the reviewer,

Kind regards,

Gino De Gelder, Navid Hedjazian, Laurent Husson, Thomas Bodin, Yannick Boucharat, Kevin Pedoja, Tubagus Solihuddin and Sri Yudawati Cahyarini

---

## Author Comment (AC2)

Dear Reviewer,

We provide hereunder an overview of how we address the comments, with explanations and references, on a point-by-point basis. Reviewers' comments are in black and our response in blue. In the manuscript, all the new/changed text is highlighted in yellow.

**Reviewer 2 Comments to Author:**

This manuscript is interesting, novel and well written, and I like it. It presents a new method that inverts the topography (mainly widths) of marine terrace staircases, simultaneously solving for sea-level history, uplift rate and some other parameters. The method seems to work pretty well on the three test datasets (one synthetic, two real).

Thank you, we appreciate the comment.

I agree with the comments of Anonymous Referee #1 — notably that for a paper focused on methodology there are quite a few details lacking. It's possible to find some information on inversion parameters in the code in the GitHub repo, but the repo is confusingly set up, with instructions focusing on how to run it on one HPC at one institution. I would recommend including discussion of the inversion parameters in the text and if possible, making the code more user friendly.

Fair point. Considering the first part, we refer to the Response_Reviewer1 document where we specified the changes made to clarify the methodology. For the second part, we improved the GitHub to make things easier to find.

I also agree that it would be good to present posterior distributions for inverted parameters: perhaps a table showing prior ranges for uplift rate, wave base height etc., together with the 95.4%-confidence ranges from the posterior distributions. It would be also nice to put some 95% contours on Figure 3e-g.

Instead of a table we opted to add the confidence ranges visually in both Figures 3 and 5.

A paragraph discussing the results of the Corinth inversion in terms of geological plausibility would strengthen the paper… Do the different posterior distributions on initial slope and erosion rate make sense given the local geomorphology of each profile swath? Relative differences in nearby sediment supply might influence erosion rates, for example.

We added a couple of lines in the text to discuss this (Lines 324-326), as it is indeed an interesting point to emphasize that erosion rate and wave base depth all overlap between the 3 profiles.

*"Although we might expect lateral differences in these rates given variability in sediment types, catchment area and coastal orientation, the broad ranges for the posterior distributions indicate we cannot quantify these lateral differences from the profile morphology alone. "*

I don't have much to say about section 6.2, but I agree with Anonymous Referee #1 that it comes as a bit of a surprise (especially given the title) and could be shortened.

We feel like the implications of the Corinth inversion are very interesting, and merit a deeper discussion beyond the methodological advances that this paper presents. We think some readers will appreciate such a discussion, and have thus decided to keep this section as detailed as it is. We did change the order of sections 6.1 and 6.2, as to end the discussion in a broader context.

To emphasize more that we find the discussion of Corinth a key part of the paper, we modified the title, which is now;

*"Bayesian reconstruction of sea-level and hydroclimates from coastal landform inversion: application to Santa Cruz (US) and Gulf of Corinth"*

and added several sentences throughout the paper to help that emphasis (Lines 76-81, 225-229, 293-296, 346-349, 355-356, 503-506).

Finally, a couple of typos:
A colon is missing between "Andersen et al., 2010" and "De Gelder" in the caption of Figure 1.

Fixed as suggested. (Line 94)

There is a "d" missing in "and" in the 4th line of page 6.

Fixed as suggested. (Line 182)

Anyway, nice paper!

Thank you!

We very much appreciate the time and effort by the reviewer to go through our manuscript, as well as the good suggestions made to improve the manuscript.

We hope the applied changes will be appreciated by the reviewer,

Kind regards,

Gino De Gelder, Navid Hedjazian, Laurent Husson, Thomas Bodin, Yannick Boucharat, Kevin Pedoja, Tubagus Solihuddin and Sri Yudawati Cahyarini

---

## Referee Report (RR1)

Comments on DeGelder et al. submitted to ESurf

De Gelder et al. developed a method to estimate past sea-level changes (on a 10 ka - 400 ka scale) from the topographic profiles of Pleistocene marine terraces through a Bayesian inversion using a topographic evolution model based on coastal erosion.

Conventionally, analyses of the relationship between marine terraces and sea-level change history have either assumed a known sea-level curve to estimate coastal uplift rates or inferred past sea-level fluctuations by first determining the formation ages of terraces. This study allows for significant flexibility in sea-level changes and uplift rates, enabling reliable estimations based solely on topographic profiles. This approach is highly innovative.

The analysis in the Gulf of Corinth, in particular, showed a remarkable achievement of the authors' method, as it enabled an independent estimation of sea-level change history in a unique environment that was not always connected to the open ocean, separate from the well-known eustatic sea-level change curves. The tectonic and geological interpretation of the estimated sea-level change history is also well-structured and logically sound.

Therefore, I have no objections to the subject, analysis, results, and interpretation presented in this manuscript being accepted in the future. However, particularly in the Methods section, the authors seem to have omitted important details and are not providing sufficient information for the readers. Below, I will point out the inadequacies in the descriptions and offer suggestions for improvement.

1. Vague descriptions in Methods

Section 2, **Marine Terrace Sequence Inversion**, describes the methods used in this study, namely the topographic evolution model as a terrace formation model and the MCMC method as a Bayesian inversion approach. In general, reports of nonlinear inversion clearly describe the following three steps to make the approach more comprehensible:

1. Mathematical formulation of the observation equation
2. Evaluation of the posterior likelihood of a proposed model
3. Implementation of the nonlinear inversion

Step (1) essentially defines what the model parameters are, what the observations are, and how they are related. A general formulation is:

$$d = f(g) + e$$

In the context of this study, $d$ represents the topographic profile, $g$ is the sea-level change curve, uplift rate, and other control parameters, and $f()$ corresponds to the REEF code. Since $f()$ is a complex nonlinear model, mathematical expression may not be necessary. Instead, it would be clearer to indicate that the observed topographic profile (e.g., $z = P(x)$) is a function of the model parameters, which can be written as $z = P(x|g)$. Also, since this study compares the horizontal positions as the observed data, an appropriate representation could be $x = P'(z|g)$.

Step (2) is the process of evaluating the likelihood by comparing the modeled observations with the true observation data. I guess the calculation the authors conducted was: When the true observed profile is represented as $x = D(z)$ and the errors are assumed to follow a Gaussian distribution with variance $\sigma^2$, then the likelihood for each elevation can be expressed as:

$$p(z|g) = \frac{1}{\sqrt{2\pi\sigma^2}} \exp\left(-\frac{\left(P'(z|g) - D(z)\right)^2}{2\sigma^2}\right)$$

and the total log-likelihood over the entire observation range is

$$\log L(g) = \log \prod p(z|g) = -\sum \frac{\left(P'(z|g) - D(z)\right)^2}{2\sigma^2} + C\,(constant)$$

This formulation assumes that errors at different elevation points are independent. However, since it is mentioned that correlations between nearby heights are considered, the authors should appropriately modify the formulation to account for these dependencies.

Regarding steps (1) and (2), I felt that the manuscript provides explanations in text that cover the necessary content. However, the explanations were not presented in a structured manner, and the lack of formal definitions for the observation data and model parameters led to unnecessary confusion.

For Step (3), since the implementation of Bayesian inversion and MCMC requires the definition of model parameters and the model likelihood evaluation, it would be more effective to introduce them after (1) and (2). While mathematical expressions would be preferable, the current explanation is likely sufficient. One suggestion is to clearly distinguish between aspects common to general Bayesian inversion and MCMC sampling and those specific to this study.

Additionally, if prior probability distributions or constraints are imposed on the model parameters, they should be described here. From the subsequent analysis, it appears that a strong prior constraint is applied to the uplift rate and sea-level curve.

2. Roles of each model parameter

The role of each model parameter in the inversion analysis does not seem to be explicitly stated. In this inversion, in addition to the sea-level history, model parameters such as IS, ER, WB, and UR are introduced. Upon first reading, I thought that both the sea-level history and the average uplift rate were being estimated as target parameters. However, it seems that the average uplift rates were pre-estimated from terrace ages and fixed in a certain range in the practical cases. While this approach is valid, it would be clearer to explicitly state that the uplift rates were treated as a known parameter and incorporated as a prior information. This would help avoid any confusion about its role in the inversion process.

Similarly, for sea-level curves, it appears that relatively strong constraints (red rectangles) are imposed in some time ranges where data are available (MIS-5e for example). However, the

manuscript does not clearly convey how much the authors weight this prior information. In other words, there are two possible roles for each node:

1. The prior knowledge from other research constrains past sea levels within a range of several tens of meters and stabilize the overall solutions.
2. No substantial constraints are imposed, and past sea levels are estimated as the solution purely from topographic data.

In the case of the Gulf of Corinth, it seems that highstands are treated as the former case (strong prior constraints), while lowstands are treated as the latter. Since the manuscript attempts to display both cases within the same framework, it may cause confusion for readers.

In the case of MIS 7a (200 ka) in the Gulf of Corinth, the strong convergence occurs near the edge of the constrained region (red rectangle). This makes it unclear whether the estimated sea level was derived from the inversion of topographic data or simply influenced by the prior constraints. In contrast, for MIS 9a and 9b, the sea level appears to converge at the center of the broadly defined prior range, demonstrating that the model itself has a strong capacity to constrain sea-level variations. If the goal is to highlight the broader applicability of this model, especially in regions where past sea-level changes are poorly constrained, applying too strong priors could lead to an underestimation of the model's capabilities.

To improve the approach, two possible alternations can be considered: (1) Removing all vertical constraints on node heights and reconstructing sea levels purely from topographic data (essentially applying the same constraint as in the synthetic test). However, if the inversion would not converge under this condition, then an alternative approach is: (2) Explicitly distinguishing highstands as constrained model parameters when they are supported by strong prior information.

In nonlinear inversion, various model parameters are incorporated, but not all of them are directly determined as the solution. Some serve as auxiliary parameters to stabilize the solution. The distinction between these roles may not be sufficiently clarified in the Methodology and Results sections.

Even in its current form, this study presents an excellent analytical approach, methodology, and results. However, a clearer articulation of what the analysis aims to resolve would elevate the research to an even higher level. If time allows, it would be valuable to examine the results when vertical constraints on sea-level heights are removed (or I guess the authors have already tried it but didn't converge within a reasonable solution). However, even without additional analysis, the current results can be interpreted as allowing some flexibility in highstand elevations, effectively treating them as soft constraints rather than fixed values.

Overall, I find the contents of this study to be highly commendable. One particularly noteworthy aspect of this approach is that joint inversion of marine terrace data that experienced the same sea-level changes but different uplift rates provides a strong constraint on the sea-level history. This study analyzed 1–3 transects per region. Was this limitation due to computational amounts or the availability of observational data? If the limitation is within

computation, it could serve as a strong motivation for future research into improving the forward algorithm, such as REEF.

Regarding the interpretation of the Gulf of Corinth results in Section 6.1, I do not have specialized knowledge of the region's tectonics and environments, so I will refrain from making definitive judgments. However, the discussion appears to be consistent with observations from previous studies, and the interpretation of the topographic evolution history is innovative. Within the context of this paper, the strong constraints placed on sea-level changes that differ from the eustatic changes of the open ocean are particularly important. I think this aligns well with the purpose of the newly developed method.

Other minor comments are below:

>L121: We use nodes interpolated through a cubic spline scheme (Fig. 2b; light blue).

Each node has values for age and elevation. The set of nodes can be expressed as $V = \{v_i = (t_i, z_i | i = 1, 2, \ldots, n)\}$

>LL132–134: ~, or left free within chosen ranges, as done for the 133 Santa Cruz and Corinth examples below.

The meaning of the analysis fundamentally depends on the size of the chosen range here: If a wide uniform prior distribution is assigned, the parameter remains flexible, and the inversion result represents one possible solution constrained by the observation. If a narrow range is imposed, the parameter effectively becomes a prior constraint. Looking at the posterior probability density distributions, it appears that IS and ER behave as free parameters, while UR acts more like a prior constraint.

>LL149–150: In this way, the terrace width is used as an observed parameter.

This sentence is highly misleading. Upon first reading, I thought that the terrace width $w$ was extracted from the topographic profile, and then the model directly outputs terrace width by some means and evaluated the misfits. However, it turned out that the horizontal distance between the modeled and observed topographic profile was used as the model error, and, as a consequence, the terrace width emerges as a primal feature.

>LL166–168: For the inversion, we fixed the nodes at 78, 6 and 0 ka, ~

Are heights also fixed?

>LL354–355: The major peaks in our reconstructed sea/lake-level curve occurred during interglacial sea-level highstands, when sea level in the Gulf of Corinth was similar to eustatic sea level (marine mode M).

Didn't the model setup constrain the ages for highstands?

>LL452–459: In the cases of Corinth and Santa Cruz we assigned relatively narrow windows for the uplift rate, ~

This paragraph addresses the concerns I had. The discussion here is well thought out, but it would be beneficial to mention also in either the model setup or the introduction that a strong prior constraint is imposed on the uplift rate, along with its meaning.

>LL470–472: It suggests that estimating paleo sea-level based on the comparison of a present-day landform to a paleo-landform (Rovere et al., 2016), may be too simplistic in many cases, at least for erosive marine terraces.

Personally, I believe that joint analysis using multiple profiles provides a very strong constraint, and I am optimistic about its potential. In Discussion, there is little mention of joint analysis. Why not emphasizing this aspect and highlight the future applicability and robustness of this approach?

>Figure 2.

It was explained that the MIS3 age is not constrained because it is currently below sea level. Wouldn't it be possible to represent this range in Fig. 2? I mean a diagonal line with the slope of UR projected from the present sea level, indicating that the triangular area below this line is submerged and not reflected in the terrestrial topography. This addition would direct the reader's attention to the upper right parts of Fig. 2b, d, and e, reinforcing the impression that the model has strong constraining power.

---

## Author Response (AR2)

Dear Editor, dear Reviewer,

We provide hereunder an overview of how we address the comments, with explanations and references, on a point-by-point basis. Reviewers' comments are in black and our response in blue. In the manuscript, all the new/changed text is highlighted in yellow.

In addition to the changes directly in relation to the Reviewer's comments, we would like to emphasize that we also added a new figure to the main text (without being asked for it). We felt like the last panel of the former Fig. 6 could benefit from a more illustrative representation to clarify the different hydroclimatic modes. As such we have made a new Fig. 7, combining the former panel with 2D sketches with 3D illustrations of the landscape. As this figure does not present new data, only a (we think) better representation of the data, and the new figure does not alter any conclusions of the paper, we hope that the Editor is ok with this new figure.

**Reviewer 3 Comments to Author:**

De Gelder et al. developed a method to estimate past sea-level changes (on a 10 ka - 400 ka scale) from the topographic profiles of Pleistocene marine terraces through a Bayesian inversion using a topographic evolution model based on coastal erosion.

Conventionally, analyses of the relationship between marine terraces and sea-level change history have either assumed a known sea-level curve to estimate coastal uplift rates or inferred past sea-level fluctuations by first determining the formation ages of terraces. This study allows for significant flexibility in sea-level changes and uplift rates, enabling reliable estimations based solely on topographic profiles. This approach is highly innovative.

The analysis in the Gulf of Corinth, in particular, showed a remarkable achievement of the authors' method, as it enabled an independent estimation of sea-level change history in a unique environment that was not always connected to the open ocean, separate from the well- known eustatic sea-level change curves. The tectonic and geological interpretation of the estimated sea-level change history is also well-structured and logically sound.

Therefore, I have no objections to the subject, analysis, results, and interpretation presented in this manuscript being accepted in the future. However, particularly in the Methods section, the authors seem to have omitted important details and are not providing sufficient information for the readers. Below, I will point out the inadequacies in the descriptions and offer suggestions for improvement.

Thank you, we really appreciate the time and effort put in to this review, and can also agree with the reviewer that some clarifications will further improve the manuscript.

1. Vague descriptions in Methods
Section 2, Marine Terrace Sequence Inversion, describes the methods used in this study, namely the topographic evolution model as a terrace formation model and the MCMC method as a Bayesian inversion approach. In general, reports of nonlinear inversion clearly describe the following three steps to make the approach more comprehensible:

1) Mathematical formulation of the observation equation

2) Evaluation of the posterior likelihood of a proposed model

3) Implementation of the nonlinear inversion

Step (1) essentially defines what the model parameters are, what the observations are, and how they are related. A general formulation is:
$d = f(g) + e$
In the context of this study, $d$ represents the topographic profile, $g$ is the sea-level change curve, uplift rate, and other control parameters, and $f()$ corresponds to the REEF code. Since $f()$ is a complex nonlinear model, mathematical expression may not be necessary. Instead, it would be clearer to indicate that the observed topographic profile (e.g., $z = P(x)$) is a function

of the model parameters, which can be written as $z = P(x|g)$. Also, since this study compares the horizontal positions as the observed data, an appropriate representation could be $x = P'(z|g)$.

We agree that the proposed distinction would make the methods section more clear, so we added some introductory sentences, and improved the description of section 2.1 following the reviewer's suggestions (Lines 99-110), that now read:

*"In the following sub-sections, we describe the inversion of marine terraces in 4 parts: 1) the unknown model parameters and their relation to observed terraces, 2) the Bayesian formulation of the inverse problem, 3) the Monte Carlo algorithm to approximate the probabilistic solution and 4) the bounds of the uniform prior ranges.*

*2.1 Model parameters*
*As a general formulation, we can consider:*
$$d = g(m) + \varepsilon \qquad (1)$$
*Where d describes the vector of observations, in our case the topographic profile of a terrace sequence, and m is the set of unknown model parameters to be inverted for: uplift rate (U), erosion rate (E\*), wave-base depth ($z_0$), initial slope ($\alpha$) and sea-level history described by a set of nodes (see below). The function g describes the numerical erosion model that links these model parameters to the topographic profile, here the REEF code (Husson et al., 2018; Pastier et al., 2019). Data errors here are given by a random variable $\varepsilon$ that describes the inability of the forward model g(m) to explain the observations d."*

We did decide to keep the mathematical expressions for the forward model as well. It was specifically requested by another reviewer, and we can imagine that other potential readers of the manuscript would also appreciate a brief description of the wave erosion model.

Note that here and throughout the manuscript, we changed the symbols for erosion rate (from ER to E\*), initial slope (from IS to $\alpha$), uplift rate (UR to U) and wave base depth (WB to $z_0$), to be consistent with the original papers describing the REEF model (Husson et al., 2018; Pastier et al., 2019)

Step (2) is the process of evaluating the likelihood by comparing the modeled observations with the true observation data. I guess the calculation the authors conducted was: When the true observed profile is represented as $x = D(z)$ and the errors are assumed to follow a Gaussian distribution with variance $\sigma 2$, then the likelihood for each elevation can be expressed as:
1 $(P'(z|g) - D(z))2$ $p(z|g) = \sqrt{2\pi\sigma2}$ exp $(- 2\sigma2 )$
and the total log-likelihood over the entire observation range is
$(P'(z|g) - D(z))2$

log $L(g)$ = log $\prod p(z|g)$ = $- \sum 2\sigma2 + C(constant)$
This formulation assumes that errors at different elevation points are independent. However, since it is mentioned that correlations between nearby heights are considered, the authors should appropriately modify the formulation to account for these dependencies.
Regarding steps (1) and (2), I felt that the manuscript provides explanations in text that cover the necessary content. However, the explanations were not presented in a structured manner, and the lack of formal definitions for the observation data and model parameters led to unnecessary confusion.

We tried to clarify this 'step 2' (section 2.2) following the reviewers suggestion. We re-arranged some text and added the appropriate equations (Lines 133-163).

For Step (3), since the implementation of Bayesian inversion and MCMC requires the definition of model parameters and the model likelihood evaluation, it would be more effective to introduce them after (1) and (2). While mathematical expressions would be preferable, the current explanation is likely sufficient. One suggestion is to clearly distinguish between aspects common to general Bayesian inversion and MCMC sampling and those specific to this study.

We re-arranged the sentences from the previous version for a better structured explanation in section 2.3 as well now (Lines 166-178).

Additionally, if prior probability distributions or constraints are imposed on the model parameters, they should be described here. From the subsequent analysis, it appears that a strong prior constraint is applied to the uplift rate and sea-level curve.

We added a sub-section about the prior constraints now (Lines 180-193):

"2.4 Bounds of the uniform prior
For the different unknown model parameters, imposed prior constraints can be either restrictive, open or anything in between. Within the synthetic tests (section 3) we left the sea-level nodes open, and the other parameters either fixed (Fig. 2) or open (Fig. S3). For the Santa Cruz benchmark tests (section 4) we left the erosion rate, wave base depth and initial slope parameters relatively open, but placed a more restrictive range on the uplift rate of 1.3-1.65 mm/yr, so that the chronostratigraphy of the modeled terrace sequence would match published ages (Perg et al., 2001). We put soft prior constraints on the sea-level history, by restricting possible solutions to the red boxes in Fig. 1c, which represents a cautious interpretation of the ensemble of previous sea-level studies. We adopted a similar strategy for the Corinth terraces (section 5), with the erosion rate, wave base depth and initial slope parameters left relatively open, but stronger prior constraints on uplift rate ranges to respect previous findings on the chronostratigraphy. We tested models with the sea-/lake-level nodes either completely open between 15 and -150 meter elevation (Fig. S4), and with stronger prior constraints only on the highstands. During these highstands the Gulf of Corinth undoubtedly experienced marine conditions (McNeill et al., 2019), so the sea-level elevations can be expected to fall within the red boxes of eustatic sea-level defined in Fig. 1c, whereas lowstands are likely lacustrine."

2. Roles of each model parameter
The role of each model parameter in the inversion analysis does not seem to be explicitly stated. In this inversion, in addition to the sea-level history, model parameters such as IS, ER, WB, and UR are introduced. Upon first reading, I thought that both the sea-level history and the average uplift rate were being estimated as target parameters. However, it seems that the average uplift rates were pre-estimated from terrace ages and fixed in a certain range in the practical cases. While this approach is valid, it would be clearer to explicitly state that the uplift rates were treated as a known parameter and incorporated as a prior information. This would help avoid any confusion about its role in the inversion process.
Similarly, for sea-level curves, it appears that relatively strong constraints (red rectangles) are imposed in some time ranges where data are available (MIS-5e for example). However, the manuscript does not clearly convey how much the authors weight this prior information. In other words, there are two possible roles for each node:

1. The prior knowledge from other research constrains past sea levels within a range of several tens of meters and stabilize the overall solutions.

2. No substantial constraints are imposed, and past sea levels are estimated as the solution purely from topographic data.

In the case of the Gulf of Corinth, it seems that highstands are treated as the former case (strong prior constraints), while lowstands are treated as the latter. Since the manuscript attempts to display both cases within the same framework, it may cause confusion for readers.
In the case of MIS 7a (200 ka) in the Gulf of Corinth, the strong convergence occurs near the edge of the constrained region (red rectangle). This makes it unclear whether the estimated sea level was derived from the inversion of topographic data or simply influenced by the prior constraints. In contrast, for MIS 9a and 9b, the sea level appears to converge at the center of the broadly defined prior range, demonstrating that the model itself has a strong capacity to constrain sea-level variations. If the goal is to highlight the broader applicability of this model, especially in regions where past sea-level changes are poorly constrained, applying too strong priors could lead to an underestimation of the model's capabilities.
To improve the approach, two possible alternations can be considered: (1) Removing all vertical constraints on node heights and reconstructing sea levels purely from topographic

data (essentially applying the same constraint as in the synthetic test). However, if the inversion would not converge under this condition, then an alternative approach is: (2) Explicitly distinguishing highstands as constrained model parameters when they are supported by strong prior information.

In nonlinear inversion, various model parameters are incorporated, but not all of them are directly determined as the solution. Some serve as auxiliary parameters to stabilize the solution. The distinction between these roles may not be sufficiently clarified in the Methodology and Results sections.

Even in its current form, this study presents an excellent analytical approach, methodology, and results. However, a clearer articulation of what the analysis aims to resolve would elevate the research to an even higher level. If time allows, it would be valuable to examine the results when vertical constraints on sea-level heights are removed (or I guess the authors have already tried it but didn't converge within a reasonable solution). However, even without additional analysis, the current results can be interpreted as allowing some flexibility in highstand elevations, effectively treating them as soft constraints rather than fixed values.

We understand the confusion, and have now clarified that in section 2.4 (see comment above). At an earlier stage we also did perform tests with the Corinth nodes left completely open, but realized the solutions would not provide us with a lot of insight within the posterior distributions. Hence the, in our eyes justified, compromise to 'guide' the high stand elevations with stronger prior constraints.
We've added those initial tests with every node free in Fig. S4 now, and apart from section 2.4, we also clarified the choice of priors in section 5 (Lines 330-333):

*"For the marine periods we follow the more restricted eustatic sea-level ranges defined in Fig. 1bc (red boxes), as the resulting posterior sea-/lake-level ranges would remain similar to the prior ranges for tests in which we also gave a lot of freedom to nodes for the marine periods (Fig. S4). "*

Overall, I find the contents of this study to be highly commendable. One particularly noteworthy aspect of this approach is that joint inversion of marine terrace data that experienced the same sea-level changes but different uplift rates provides a strong constraint on the sea-level history. This study analyzed 1–3 transects per region. Was this limitation due to computational amounts or the availability of observational data? If the limitation is within computation, it could serve as a strong motivation for future research into improving the forward algorithm, such as REEF.

The limited amount of transects was mostly to keep things simple, and the main purpose of this paper was to test the model on increasingly complex settings. We clarified this now in the discussion (LInes 507-508): *"For simplicity, as the aim of this paper was to test the model on increasingly complex settings, we only inverted 1 profile for Santa Cruz and 3 for Corinth, but it is easily possible to explore with more profiles at those locations in future work."*

Regarding the interpretation of the Gulf of Corinth results in Section 6.1, I do not have specialized knowledge of the region's tectonics and environments, so I will refrain from making definitive judgments. However, the discussion appears to be consistent with observations from previous studies, and the interpretation of the topographic evolution history is innovative. Within the context of this paper, the strong constraints placed on sea- level changes that differ from the eustatic changes of the open ocean are particularly important. I think this aligns well with the purpose of the newly developed method.

We appreciate the comment, thank you.

Other minor comments are below:

>L121: We use nodes interpolated through a cubic spline scheme (Fig. 2b; light blue). Each node has values for age and elevation. The set of nodes can be expressed as $V = \{v_i = (t_i, z_i | i = 1, 2, ..., n)\}$

Changed as suggested (Lines 126-128):

*"We use nodes interpolated through a cubic spline scheme (Fig. 2b; light blue), in which each node has values for age and elevation, and the set of nodes can be expressed as:*
$$v_i = (t_i, z_i \,|\, i = 1,2,...,n) \qquad (4)"$$

>LL132–134: ~, or left free within chosen ranges, as done for the 133 Santa Cruz and Corinth examples below.
The meaning of the analysis fundamentally depends on the size of the chosen range here: If a wide uniform prior distribution is assigned, the parameter remains flexible, and the inversion result represents one possible solution constrained by the observation. If a narrow range is imposed, the parameter effectively becomes a prior constraint. Looking at the posterior probability density distributions, it appears that IS and ER behave as free parameters, while UR acts more like a prior constraint.

We hope to have clarified this now with the method section on prior ranges (Lines 180-193).

>LL149–150: In this way, the terrace width is used as an observed parameter.
This sentence is highly misleading. Upon first reading, I thought that the terrace width $w$ was extracted from the topographic profile, and then the model directly outputs terrace width by some means and evaluated the misfits. However, it turned out that the horizontal distance between the modeled and observed topographic profile was used as the model error, and, as a consequence, the terrace width emerges as a primal feature.

We agree this sentence was misleading, and have removed it

>LL166–168: For the inversion, we fixed the nodes at 78, 6 and 0 ka, ~ Are heights also fixed?

We clarified this now: *"we fixed the elevation and ages of the nodes at 0, 0 and -30 m and 78, 6 and 0 ka, respectively,"* (Lines 207-208)

>LL354–355: The major peaks in our reconstructed sea/lake-level curve occurred during interglacial sea-level highstands, when sea level in the Gulf of Corinth was similar to eustatic sea level (marine mode M).
Didn't the model setup constrain the ages for highstands?

Yes it did, so it is not a surprise, but for the structure and flow of this section we wanted to mention it anyway. We have clarified *"as direct consequence of our choices in the ranges of priors,"* (Lines 379-380).

>LL452–459: In the cases of Corinth and Santa Cruz we assigned relatively narrow windows for the uplift rate, ~
This paragraph addresses the concerns I had. The discussion here is well thought out, but it would be beneficial to mention also in either the model setup or the introduction that a strong prior constraint is imposed on the uplift rate, along with its meaning.

Fair enough, we have clarified that within the method section on prior ranges now (Lines 180-193).

>LL470–472: It suggests that estimating paleo sea-level based on the comparison of a present-day landform to a paleo-landform (Rovere et al., 2016), may be too simplistic in many cases, at least for erosive marine terraces.
Personally, I believe that joint analysis using multiple profiles provides a very strong constraint, and I am optimistic about its potential. In Discussion, there is little mention of joint analysis. Why not emphasizing this aspect and highlight the future applicability and robustness of this approach?

We did mention the joint analysis in the following paragraph already: *"One of our key findings is that inverting multiple profiles simultaneously provides much better paleo sea-level constraints than focusing on individual profiles (Figs. 2, 5)."* (Lines 511-512), but have now also emphasized it in this paragraph: *"To reduce uncertainties, we demonstrated in this paper that a joint inversion of multiple profiles is a powerful way forward."* (Lines 505-506)

>Figure 2.
It was explained that the MIS3 age is not constrained because it is currently below sea level. Wouldn't it be possible to represent this range in Fig. 2? I mean a diagonal line with the slope of UR projected from the present sea level, indicating that the triangular area below this line is submerged and not reflected in the terrestrial topography. This addition would direct the reader's attention to the upper right parts of Fig. 2b, d, and e, reinforcing the impression that the model has strong constraining power.

We understand the idea, although the reviewer probably refers to the MIS1 terrace (not MIS3). We considered adding such a line, but find it to be complicating the figure rather than clarifying. As mentioned in Lines 223-225:

"*Although the MIS 1 terrace is not inverted, there are some limitations to the magnitude and rate of sea-level rise between MIS 2 and MIS 1 (Fig. 2b), probably because this period determines how much of the MIS 3 terrace is eroded at its distal edge.*"

So to some extent the sea-level changes below such a diagonal line, from 0 m at 0 ka as proposed by the reviewer, would still have an impact on the terrestrial topography.

We very much appreciate the time and effort by the reviewer to go through our manuscript, as well as the good suggestions made to improve the manuscript.

We hope the applied changes will be appreciated,

Kind regards,

Gino De Gelder, Navid Hedjazian, Laurent Husson, Thomas Bodin, Yannick Boucharat, Kevin Pedoja, Tubagus Solihuddin and Sri Yudawati Cahyarini

---

## Author Response (AR3)

Dear editor,

We thank you for your time and effort to go through our manuscript once again. Concerning the last remaining reviewer remarks:

*1. Please ensure that all vector variables (e.g., d, m, ε) are in bold italics.*

We have done that throughout the text

2. LL 138–149: The paragraph describing the comparison between observation and modeled values is still somewhat wordy. A more mathematical presentation may help clarify the explanation.
For example, it may help to define the observed profile as d(z) and the predicted profile as m(z), so that the observation equation can be written as $d_i = m_i + ε_i$, where $d_i = d(z_i)$, $m_i = m(z_i)$, and i = 1, 2, ..., N.

We have shortened that paragraph, and placed in the last paragraph of section 2.2 now. It reads:

"In this work, the data vector is defined as the horizontal position of a set of points measured on the shoreline with a vertical step size (*ipstep*; Fig. 2a). That is, the observed profile is defined as *d($z_i$),* where $z_i$ (with *i=1,2,..,N*) is a regular grid of elevations.  The misfit between this observed topography and the modelled paleo-shoreline sequences is therefore measured by comparing the horizontal distance between observed and simulated horizontal position at each vertical grid point $z_i$."

3. I find the new Figure 7 to be very helpful for understanding the interpretation. It would be even better if a north arrow could be added to the map for orientation.

We have added the north arrow

We are looking forward to seeing this manuscript published soon, best regards,

Gino de Gelder, Navid Hedjazian, Laurent Husson, Thomas Bodin, Anne-Morwenn Pastier, Yannick Boucharat, Kevin Pedoja, Tubagus Solihuddin and Sri Yudawati Cahyarini